# Hierarchy-Dependent Behaviour of Dogs in the Strange Situation Test: High-Ranking Dogs Show Less Stress and Behave Less Friendly with a Stranger in the Presence of Their Owner

**DOI:** 10.3390/ani15131916

**Published:** 2025-06-28

**Authors:** Viktória Bakos, Kata Vékony, Péter Pongrácz

**Affiliations:** 1Department of Ethology, ELTE Eötvös Loránd University, Pázmány Péter sétány 1/c, 1117 Budapest, Hungary; bakosvikee@gmail.com (V.B.); kata.vekony.kami@gmail.com (K.V.); 2Institute of Cognitive Neuroscience and Psychology, HUN-REN Research Centre for Natural Sciences, Magyar Tudósok krt. 2, 1117 Budapest, Hungary

**Keywords:** dog–owner attachment, hierarchy, rank score, strange situation test, cohabiting dogs

## Abstract

The attachment bond between companion dogs and their owners is one of the main synchronizing factors in dog–human coexistence. Attachment is based on the asymmetric dependence of the dog on the human partner. Dogs seek the owner’s proximity when they experience threats, and more readily explore novel stimuli when their owner is present. There can be differences in the finer details of attachment in dogs; however, so far, no main background factors have been described in association with these differences. In this research, we tested whether dogs from multi-dog households would show different attachment behaviour as a function of their relative position in the hierarchy among the other cohabiting dogs. We found that when their owner was present, dogs with higher rank scores were less friendly with the experimenter in the strange situation test and showed, in general, less stress signs than dogs with lower rank scores. Interestingly, the older a dog was, the fewer stress-related behaviours they showed; however, these dogs were friendlier with the stranger. These are the first results showing that dogs’ rank may affect the attachment bond with the owner, potentially influenced by the outcome of cohabiting dogs’ competition for the owner as an undividable resource.

## 1. Introduction

Man’s best friend—probably everyone has heard this colloquial description of dogs, which refers not only to the ubiquitous usefulness of the oldest domesticated species [1], but also puts an emphasis onto the extremely sociable nature of this animal with humans [2]. Researchers realized since the late 1990s that the natural habitat for dogs is the anthropogenic niche [3]. Consequently, since their domestication, the main species-specific socio-cognitive and behavioural features of dogs were formed under such pressures of selection that resulted in high-level human compatibility in dogs [4]. Although approximately 75% of the world’s dog population can be considered free-ranging (often dubbed as ‘pariah’, or ‘village’) dogs [5], even these animals can be considered socialized, and they show behavioural skills [6] that help them to coexist with humans without major conflicts.

Humans and dogs coexist in an asymmetrical relationship: dogs are dependent on human resources, since with domestication they evolved from the ancestral cooperative apex predator to a scavenger [7,8]. Therefore, dogs show a wide range of socio-cognitive and behavioural skills that signify their dependency on humans [9] that secure their reliable access to such essential resources as food and shelter (owned and ownerless dogs alike [10,11]), and even unconditional human caretaking [12]. Compared to wolves (the closest wild relative of dogs), dogs became more competitive and less cooperative with their conspecifics; meanwhile, they readily cooperate with humans [13]. This shows that for dogs, the possible closest bond with humans became a pivotal feature from the aspect of survival—but the question emerges at the same time: are dogs equally skillful in their essential bond-building with humans? What would be the factors that could cause differences in dogs’ human-directed behaviours?

Among the most decisive behavioural and socio-cognitive capacities that enable dogs to become an easily accepted and successful part of the human social system, we can mention their communicative behaviours (both as receivers—[14] and signalers—[15]); their reliance on human intervention in difficult situations (‘unsolvable task’—[16]); preference for behavioural solutions that were obtained through observational learning from human demonstrators [17]; and their sensitivity to human intentions (marked by ostensive communication [18]) and emotional states [19]. These features can all be considered as components of the dogs’ system of dependency on humans, because they develop through spontaneous interactions with humans without apparent training, and they enable dogs to adhere to human habits, guidance, and signals, which makes dogs a pleasant, interactive companion and useful working animal. However, there is considerable variability among dogs in the case of these aforementioned behaviours. Training and lifetime experiences with humans can affect their success in following human communicative signals [20] and their reliance on human support in the ‘unsolvable task’ [21]. Functional breed selection (i.e., whether a breed was selected for working independently, or in a close cooperation with the handler) affected dogs’ performance in following the visual signals of humans [22] and their performance in a social learning task [23]. However, these genetic and experience-based differences can all be explained by such circumstances, where humans needed dogs that were trained or selected for activities or tasks with various amounts of emphasis on/usefulness of the involved skills of the dog. There is one fundamental behavioural trait in dogs that so far seems unaffected by any of the commonly investigated influencing factors: their attachment to the owner.

Attachment in dogs is another (probably the most evident) component of their dependency on humans [24]. Dogs form an exclusive attachment bond with their owner, which remains unchanged throughout their adulthood, as long as they live with the same owner. However, even adult dogs can establish attachment with a new owner in case the previous one is lost [25]. The dog–owner attachment is considered a special analogy of the bond between a human infant and their caregiver [26], as it can be characterized with similar asymmetry between the roles of the two parties. The ‘attachment figure’ provides safety and security to the attached (dependent) partner in the case of challenging (stressful) situations. The standard testing method for the attachment bond is the strange situation test (SST), where the original version has been developed for human infants [27], but later, it was successfully adapted for dogs and their owners as well [28]. Consecutive episodes of the SST generate mild but constantly increasing stress in the infant or dog, because the subjects are taken to a previously unknown place, and in the episodes, they encounter a friendly stranger; meanwhile, they are left alone by the caregiver/owner in some of these episodes. At the final episode, the subject is completely left alone for a short time. Importantly, the SST activates two fundamental mechanisms in the subjects: ‘Safe heaven’ means that in the case of a stressful stimulus, the dependent subject will seek comfort in the vicinity of the caregiver. ‘Secure base’, on the other hand, ensures that in the presence of the attachment figure, the subject would more confidently explore the environment. The main feature of the exclusive attachment bond between the dog and its owner is that neither the safe heaven nor the secure base role of the owner cannot be substituted by anybody else [28].

Similarly, as in the case of human infants [29], in some investigations it was found that the behaviour of dogs can also be sorted into different clusters, or ‘styles’, of attachment with their owner [30]. These clusters were based on three main factors, or behavioural components, of the bonding complex: the acceptance (of the stranger) dimension, anxiety dimension, and attachment dimension [28]. However, after almost three decades of research, we still do not have any conclusive insight into which (extrinsic, intrinsic, or both) factors could be responsible for differences in the dog–owner attachment bond. There are some negative results so far; for example, it was found that dog breeds that were clustered according to their original working task (cooperative vs. independent working dog breeds) did not differ in their attachment to the owner [31]. Recently it was also shown that even the free-ranging dogs, adopted from the streets of India, can become well-functioning companions that show strong ‘attachment’ to their owner (according to a questionnaire survey [32]). Based on the fundamental role attachment plays in dogs’ lives, and its apparently uniform presence among owned dogs, one could hypothesize that this trait has become largely resilient to environmental influences, and at the same time, dogs were probably shaped by natural and artificial pressures of selection to be equally able in forming well-functioning attachment bonds with their owners.

Dogs can establish hierarchies among themselves when they live in groups—and not only the free-ranging dogs that often have to compete for basic resources [33], but also the companion dogs in dog daycare [34] and dogs from the same households [35]. The rank of a dog can affect its behaviour in resource competition (e.g., obtaining and possessing a reward [35]), or in otherwise competitive situations (e.g., leading the group [36]). Moreover, it also shows association with seemingly non-competitive features, such as social learning performance [37] or the dog’s personality traits [38]. According to our recently formulated hypothesis, companion dogs in multi-dog households have no more important ‘resource’ to compete for than the owner’s attention, vicinity, and interactions [38]. Based on this, one could assume that cohabiting dogs who hold different ranks in the domestic hierarchy may react differently to the mildly stressful strange situation test, because this protocol relies on their owner-directed bonds and relationship (‘safe heaven’ and ‘secure base’).

### Hypotheses and Predictions

In this study, we hypothesized that dogs’ rank will show association with their attachment complex-related behaviours, as measured in the SST. We predicted that higher-ranking dogs would show stronger signs of stress and attachment in the test because, being time after time separated from their owner, it could be the sign of a potential threat to their exclusive access to the most important resource. We also predicted that lower-ranking dogs would behave more friendly with the stranger (unknown person) as they realized their secondary position in the competition for the owner. To find out the answer to these predictions, in our research, we tested cohabiting dogs with their owners in the standard SST protocol, and after establishing their relative ranks by using a questionnaire (‘DRA-Q’—Dog Rank Assessment Questionnaire [35,38]), we analyzed, whether the dogs’ behaviour shows associations with their rank.

## 2. Materials and Methods

The research involved two parts: the above-mentioned questionnaire to assess dogs’ ranks in the household hierarchy and the strange situation test.

### 2.1. Dog Participants

We tested 62 dogs (*n* = 62; mean age: 6.19 years; *n* = 33 males; and *n* = 29 females) living with at least 1 other dog at home for minimum six months a priori to the testing. Dogs came from 31 households, of which 26 were two-dog, 3 were three-dog, and 2 were five-dog homes. No restrictions were placed on reproductive status, breed, or sex to participate, because previously we found [35] that the dogs’ rank score showed no significant association with the sex and reproductive status of the subjects. The other reason why we did not balance the sample of participating dogs according to their sex and reproductive status was the difficulty in finding enough multi-dog households where the cohabiting dogs’ ages were suitable for testing and the owners were willing to bring both dogs for testing to our laboratory. However, it was a requirement that the dogs did not visit the testing facility at the Department of Ethology at least a year preceding the behavioural test, and that they had not been tested in the exact same experimental room ever. We requested the owners not to bring any dog to the tests who showed severe signs of separation-related problems. Participation was voluntary, and we recruited the owners of companion dogs through social media advertisements.

### 2.2. Questionnaire

We used the DRA-Q questionnaire, developed earlier for the assessment of dogs’ rank in multi-dog households [35] and designed for owners with multiple dogs at home. Owners had to complete the questionnaire before they took their dogs for testing to the Department of Ethology. The main owner of the dogs was always requested to complete the questionnaire, who then also took part in the behavioural test with the dog (see below). In addition to the demographic data of the dogs, the instrument contained eight questions about the behaviour of the dogs in everyday situations that were related to hierarchy. Each question was scored on a scale of 1 (indicating dominant behaviour), −1 (indicating subordinate behaviour), or 0 (suggesting similar behaviour exhibited by the dogs, depending on the context). Averaging the responses across the eight questions yielded a mean rank score for each dog within each household, ranging from +1 to −1. Questions where owners indicated that a specific behaviour or situation never occurred in the case of their dogs were omitted from the rank score calculation for those dogs (Table 1). Importantly, the questionnaire was completed for all dogs who participated in the experiment; thus, each dog had a rank score, and based on this, it could also be assessed in the individual households which dog had the higher or lower rank.

### 2.3. Strange Situation Test

#### 2.3.1. Experimental Setup

Originally the test was developed to study the attachment bond of human infants to their mothers [39], and it was later adapted by Topál and colleagues to examine the relationship between companion dogs and their owners [28]. The test consists of six episodes during which the dog is exposed to increasing levels of stress induced by factors such as the unfamiliar environment, the presence of a stranger in certain episodes, and the absence of the owner in others. Lenkei and colleagues made some adjustments to the protocol because they noticed that average companion dogs showed less signs of stress nowadays than 25 years earlier. The current experiment protocol is based on the paper of Lenkei et al. [31]. According to this, to ensure the necessary level of stress during the test, we played an 8 s-long “bone-guarding” growl [40] to the dogs during the warm-up phase of the test [31]. This additional stressor became necessary after more than 20 years of the original study on dogs’ attachment behaviour by Topál and colleagues [28], when researchers noticed that companion dogs did not show the optimal low-level stress in the SST anymore [31]. The assumed reason for this change in dogs’ behaviour was the recent increased level of exposure to unknown places, persons, and situations in the average companion dogs.

Subjects were tested in an unfamiliar room (6.27 m × 5.40 m, Figure 1). The room had two doors, one for the owner and the other for the stranger during the test. Two chairs were placed facing each other approximately in the middle of the room. In one corner of the room, a crate was placed with a speaker inside (JBL Flip 5), covered with a blanket. Within one meter besides the cage, two balls and two sticks were placed on the floor, which could be used as toys during certain episodes of the test. Two tables were also placed in the room with six wooden toy blocks placed on one of them. Before the test, the owners were provided with wireless headphones through which they received instructions on what actions to take (e.g., interact with the dog, sit on the chair) with the help of a pre-recorded set of standard instructions. The tests were recorded with six cameras (Basler a2A1920-51gcPRO-Basler ace 2, Basler AG, Ahrensburg, Germany; microphone: Sennheiser ME-64 + K6-P power module, Sennheiser Electronic GmbH & Co., Wedemark, Germany; and audio interface: Focusrite-Scarlett 2i2, Focusrite PLC, Wycombe, UK) mounted on the ceiling to cover the entire room.

#### 2.3.2. Experimental Procedure

##### Warm-Up Phase (1 min)

Before the test, the experimenter (E) instructed the owner (O) about the details of the task, then led them into the room where they let the dog explore freely, always using the stranger’s door to enter and leave the room. They walked around the room twice, introducing it to the owner. When the dog approached the crate, the experimenter played a growling sound from their phone via Bluetooth on the speaker in the crate. If the dog did not go close enough to the covered cage, the owner and E stood near it to facilitate the dog’s approach, and if it still did not come closer, the growling was played after 50 s. After a minute, the owner put the dog on leash again, and they left together through the stranger’s door.

##### Testing Phase (12 min)

The experimenter guided the owner back into the room, instructed them to sit, and asked the owner to leave the leash on the chair throughout the subsequent episodes of the test. E started to play the recording with the instructions and started measuring the time with a stopwatch (later used also by the stranger to follow the phases) and left the room. The 12 min test comprised six phases, each two minutes long, with different actions performed every 30 s (Table 2). During this, the dog was in the testing room all along, sometimes with the owner, or the stranger, with both, or alone, while both the stranger and owner performed the following actions:Sit on the chair: the owner/stranger did not initiate interaction with the dog, but responded if the dog initiated the interaction (e.g., by placing its head on the owner’s knee) when it was allowed for the person to briefly pet the dog (i.e., a few hand-strokes to the head).Cube carrying: the owner/stranger carried the wooden building blocks from one table to another, completely ignoring the dog.Playing with the dog: The owner/stranger played with the dog using the available toys in the room in a natural manner. If the dog did not want to play, the owner/stranger petted the dog instead.Leave the room: the owner/stranger left the room without saying anything to the dog.Enter the room: After entering the room, the owner/stranger paused beside the door for 5 s. If the dog approached immediately, they greeted and petted the dog. If the dog did not approach, they verbally greeted the dog, waited for 5 s, then took a seat.

### 2.4. Behavioural Coding

The test videos were scored using Solomon coder (Version beta 17.03.22 © András Péter, free software, (RRID:SCR_016041). During the analysis, we used the exact same scoring system developed by Kovács et al. [41]. Three distinct scoring dimensions were established (Table 3). The first assessed ‘attachment’ (to the owner). The behavioural elements in this dimension refer to the proximity-seeking of the dog: it remains in the vicinity of the owner, follows their activity within the room, escorts the owner to the door when they leave, and greets the owner when they return. Importantly, when the owner is absent from the room, the dog does not play with the stranger, but vocalizes, orients towards or stands in the door where the owner left, and stays near the chair where the owner was sitting. The second dimension is called ‘anxiety’, referring to the distress expressions of the dog induced by the unfamiliar environment. The behavioural elements in this dimension show that the dog behaves anxiously even when the owner is in the room (the dog attempts to leave the room, does not play with the owner). Additionally, when the owner is absent, the dog may follow even the stranger when they leave the room. The third dimension (‘acceptance’) measures the quality of interactions with the stranger. The behavioural elements show that the dog plays with the stranger independently of the presence of the owner, approaches the stranger, and initiates contact with them. This scoring system involved subtracting certain behaviour scores from the total instead of adding them. For instance, in the anxiety dimension, behaviours indicating calmness in the absence of the owner (variables Calm1 and Calm2) were scored negatively. However, during data analysis, dimensions where negative scores were possible were shifted into positive ranges. Thus, the maximum score for attachment dimension was 11 points, for the anxiety dimension it was 13 points, and for the acceptance dimension it was 12 points (Table 3). Twenty percent of the videos were scored by a coder who was blind to the experiment’s hypotheses for reliability. Intercoder reliability for all three test scores suggested moderate to excellent consistency (0.94 for the attachment dimension, 0.90 for the anxiety dimension, and 0.79 for the acceptance dimension). We used inter-class correlation from the irr package in R.

### 2.5. Statistical Analysis

The data were analyzed by using R statistical software (Version v4.3.0, R Development Core Team, 2021) in RStudio (RStudio Team, Boston, MA, USA) with packages DataExplorer, emmeans, glmmTMB, lme4, lmerTest, MuMIn, outliers, and performance.

We used generalized linear mixed models with test scores as dependent variables, and questionnaire scores and demographic data (age, sex, and reproductive status) as predictor variables. We tested at least two dogs from each household. We used the ID of the shared household as a random factor to account for the cohabiting dogs’ rank scores’ non-independence, as well as their shared environment. We used an AIC-based model selection to find the parsimonious model.

## 3. Results

### 3.1. Attachment Dimension

The attachment dimension only had a trend-like negative association with the age of the dog (β = −0.0216, SE = 0.01, t = −1.956, 95% CI = (−0.0433–0.0004), and *p* = 0.051), as older dogs showed less attachment behaviours. Examining episodes with the owner present and absent separately revealed that this trend-level association can only be found when the owner is present (β = −0.09938, SE = 0.05, t = −1.876, 95% CI = (−0.2032–0.0045), and *p* = 0.061). We found no association between any of the variables and attachment dimension when the owner was absent.

### 3.2. Anxiety Dimension

When the owner was present, we found a negative association between the anxiety dimension and rank score (β = −0.1406, SE = 0.07, t = −2.049, 95% CI = (−0.2749–−0.0062), and *p* = 0.045, Figure 2). However, we found no significant association between the anxiety dimension and age or reproductive status.

When the owner was absent, we found associations between the anxiety dimension and age and reproductive status. Although the association with age was only a trend (β = −0.1006, SE = 0.05, z = −1.851, 95% CI = (−0.2071–0.0059), and *p* = 0.064), reproductive status remained significant (β = 1.2747, SE = 0.43, z = 2.948, 95% CI = (0.4272–2.1222), and *p* = 0.003).

The anxiety dimension showed significant association with dogs’ age (β = −0.1880, SE = 0.08, t = −2.365, 95% CI = (−0.3425–−0.0328), and *p* = 0.021), where older dogs were less anxious throughout the test. Reproductive status also had a significant association with the anxiety dimension: post-hoc test revealed that neutered dogs were more anxious than intact ones (β = −1.2710, SE = 0.59, t = −2.173, 95% CI = (0.1600–2.3799), and *p* = 0.034).

### 3.3. Acceptance Dimension

The acceptance dimension (consisting of behaviours related to accepting the stranger) had a significant association with age (β = 0.2219, SE = 0.10, t = 2.125, 95% CI = (0.0112–0.4269), and *p* = 0.038), and older dogs showed more acceptance toward the stranger. Rank score also had a significant, but negative association with the acceptance dimension (β = −1.4933, SE = 0.61, t = −2.468, 95% CI = (−2.6760–−0.3005), and *p* = 0.02): dogs with higher rank score accepted the stranger less.

We found similar associations between the acceptance dimension and age and rank score in the episodes when the owner was present (age: β = 0.1369, SE = 0.04, t = 3.073, 95% CI = (0.0479–0.2250), and *p* = 0.003; rank score: β = −0.6018, SE = 0.29, t = −2.107, 95% CI = (−1.1621–−0.0401), and *p* = 0.042). However, this was the only analysis where the model using the formal, agonistic, and leadership subscores of the rank score had a better fit (∆AIC = 3.03, *p* = 0.025). We found a significant association between the acceptance dimension and age (β = 0.1461, SE = 0.04, t = 3.401, 95% CI = (0.0609–0.2305), and *p* = 0.001), sex (β = 0.5355, SE = 0.26, t = 2.088, 95% CI = (−1.0233–−0.0363), and *p* = 0.042), and the leadership subscore (β = −0.5911, SE = 0.21, t = −2.776, 95% CI = (−1.0048–−0.1778), and *p* = 0.009, Figure 3): older dogs, female dogs, and dogs with lower leadership scores accepted the stranger more easily.

## 4. Discussion

During our research, we aimed to answer the question of whether social rank in the group hierarchy of dogs affects some dimensions of their attachment to the owner. We used a behaviourally validated questionnaire to determine the dogs’ rank score, which reflects the steepness of the hierarchy and different aspects of dominance. To assess attachment, we used a well-established behaviour test, called the strange situation test (SST). We found several associations between rank score and the dimensions of the attachment complex. Dogs with higher rank scores showed less stress-related behaviours. These dogs stressed less in the owners’ presence (but not in absence) than the ones with lower rank scores. We also found a negative association between rank score and the acceptance of the stranger. Dogs with higher rank scores showed fewer friendly behaviours towards the stranger during the test. Moreover, we found some non-rank-related associations. Older dogs showed less stress during the test, and they were friendlier with the stranger than the younger ones. The opposite effect of rank and age regarding dogs’ reaction towards the stranger is especially interesting, because previous studies found a positive association between age and rank score [38]. Our current result from the SST indicates that the higher-ranking dogs’ more negative reaction to the stranger could be in direct association with their rank, without the confounding influence of their age.

Here, we should emphasize that throughout our research, we considered dogs’ rank score as a relatively stable attribute of the cohabiting dogs at the time of the testing. Although the position of dogs in the hierarchy is secured through dynamic interactions, and the ‘holistic’ rank consists of various components (‘agonistic’, ‘formal’, and ‘leadership’), the multi-question DRA-Q instrument provides the opportunity for a reliable and biologically relevant assessment of the structure of an established rank system among cohabiting dogs [35]. While environmental and demographic factors may influence the changes in the individual dog’s rank, in our investigation, we tested adult subjects who a spent long enough time in mutual cohabitation for the development of stable rank conditions. The complex interrelationship between dogs’ rank and their personality traits [38] could also provide relative stability to dogs’ position in the hierarchy, protecting it from temporary fluctuations in strength and motivation, caused by, for example, sickness or environmental stress.

Originally, we hypothesized that dogs’ rank will show an association with their behaviour in the SST, where we predicted that higher-ranking dogs will show stronger attachment and higher stress levels, while lower-ranking dogs will be more friendly toward the stranger. Our main hypothesis was partially confirmed by the results. Although two of the three components of the dogs’ attachment complex (anxiety and acceptance dimensions) were associated with the dogs’ rank, the ‘attachment dimension’ itself did not differ between the high- and low-ranking dogs. This result confirms that the attachment complex towards the owner could be one of the most stable features of the dog–human relationship. It has been found that the capacity to develop an attachment bond with the owner remained unaffected by the relatively recent functional breed selection [31]. Here, we found that the attachment dimension also stays stable when dogs establish their position in the hierarchy through longitudinal (lifetime) interactions with each other. Dependence on humans is one of the most important and indispensable attributes in the companion dogs’ life [10,42]. The evolutionary and everyday success of the dog depends on a rich inventory of synchronizing mechanisms (e.g., attachment [28]; various forms of communication (e.g., [14]; social learning [17]; cooperation [8])), where dependency on humans can be regarded as a main organizing factor. In the case of permanently ownerless dogs (such as the free-ranging dogs in India), a flexible and quickly adjustable level of dependence was found on human-provided food, which was mostly influenced by the valence in the human’s interactions with dogs [6,43]. In contrast to the free-ranging dogs, for dogs who belong to a particular person (‘owner’), the best way to secure the essential resources is to develop an individualized and permanent bond to the owner. This can explain our findings when the attachment dimension was found to be fundamentally independent of the rank of the cohabiting dogs.

We predicted that high-ranking dogs would show stronger stress reactions during the SST, which includes episodes where the owner leaves the dog either alone or in the company of an unknown person. However, we found that while higher-ranking dogs showed fewer stress signs, lower-ranking dogs stressed more, which at first sight contradicts our prediction. We should note that these differences were only noticeable when the owner was present, which still highlights the specific importance of the owner as a stress reducer for the higher-ranking dog. This result supports our assumption that the dominant dogs may rely more on their owner than the subordinate dogs, and the presence of the owner provides more reassurance to the higher-ranking dogs against stress than it does to lower-ranking dogs. Dominant dogs’ stronger connection with, and reliance on humans was also found in other contexts, for example, in such frustration-eliciting situations where a human partner withheld the reward from the dogs. Dominant dogs (with a high ‘leadership’ score) behaved more demandingly with the human in these situations, showing that they rely more on interacting with humans than subordinate dogs do [44]. In the same study, it was also found that subordinate dogs tried to obtain the unreachable food in a more independent manner, without relying on the help from the human partners, while higher-ranking dogs more likely stared at their owner when the food was locked within a cage [44]. Subordinate dogs showed less human-dependent behaviours in social learning tasks. They benefitted less from observing either a stranger or familiar demonstrator in the detour task than the dominant dogs did [37,45]. Based on these facts, it can be emphasized that the presence of the owner will not relieve them in a stressful situation.

We found that when the owner was present, dominant dogs were accepting the unknown person (‘stranger’) less than subordinate dogs did. This result aligns well with our theory that the owner represents the main ‘resource’ for dogs [35,38]. We can assume that since dominant dogs could secure primary access to the owner, they are less likely to initiate amicable interactions with other people when their owner is accessible. Subordinate dogs on the other hand, being in a less favourable position in the competition for the owner, are probably more open towards interacting with other humans. These results can have relevance for such earlier findings, when some owned dogs were found to be more interested towards strangers than towards their owner in a familiar context [46]. Furthermore, it is an intriguing question whether subordinate dogs would accept more easily a new owner when rehoming becomes necessary than the higher-ranking dogs do.

We should not forget that the above-mentioned associations between dogs’ rank and the components of an attachment complex manifested themselves only in the presence or the owner. On one hand, this indirectly implies that being separated from the owner for a short time is equally stressful for both the dominant and the subordinate dog. On the other hand, while the presence of the owner may be an effective stress releaser for the dominant dogs, subordinate dogs might seek comfort with an unknown person in mildly stressful situations. In the future, it would be interesting to investigate whether dominant or subordinate dogs react differently to shorter or longer separation from the owner. It was found that dependence on humans on the breed-type level had an association with separation-related problems (cooperative working dogs showed higher levels of stress in a short separation episode than the independently working breeds did) [47]. One could expect that higher-ranking dogs, who show higher owner dependence, may manifest more frequent separation-related stress signs as well.

Another interesting result is the association between dogs’ age and their acceptance of the stranger. Older dogs were more amicable with the unknown person, and at the same time, we know that dogs’ rank and their age show a positive association with each other [38]. Dogs with higher rank scores accepted the stranger less than subordinate dogs did, thus dogs’ age and their rank acted differently in this case. This result aligns well with the associations between personality traits and dogs’ rank, where earlier it was found that most of these (openness, extraversion, and conscientiousness from the canine ‘Big Five’ inventory) were associated differently with dogs’ rank than as we would expect if age would act as a confounder [38].

The rank- and age-related findings in our study can be the result of the involvement of young, high-ranking dogs and older, low-ranking dogs among the tested subjects, who eventually could have a stronger cumulative influence on the stranger-acceptance results. Another explanation is that we did not make a global binary categorization across the subjects (dominant or subordinate) when determining the social rank. We used a scoring method, and the points were located on a scale. As a result, it is possible that young dominant dogs have higher rank scores than the older dominant individuals had. In addition, our subjects were very diverse regarding both their age and their rank scores. One could expect that as dogs grow older, they become more accepting towards strangers, even if they are otherwise higher-ranking individuals. At the same time, the low-ranking old dogs would still show the highest acceptance towards strangers.

## 5. Limitations and Future Directions

In this study, we did not analyze the various attachment styles of companion dogs (e.g., [28]). Although the participating dogs did not have separation problems according to the owners, they were not assessed by a certified veterinary specialist before the testing. As separation problems were found to be associated with particular styles of attachment [48,49], the lack of objective knowledge about our participants’ reaction to separation should be taken as a limitation.

Among the limitations, we should keep in mind that although the DRA-Q instrument showed consistent results with the outcome of such behavioural tests that showcased competitive and non-competitive interactions between cohabiting dogs [35], questionnaires that are based on the owners’ responses can always be the source of a biased assessment of rank. Furthermore, the DRA-Q contains only one item that is connected to the ’formal dominance’ component, which weakens the conclusions regarding this aspect of dogs’ rank. It must be also mentioned that the differences in the rank score between the cohabiting dogs from the same household were not equally large everywhere. This could weaken the contrast in their behavioural responses in the SST. The study could have a higher sample size as well. However, finding owners who had at least two suitable cohabiting dogs available for testing was a strong limiting factor, for example, because none of the dogs who earlier were tested in our testing room could participate in this experiment.

Our study is the first that showed associations between the rank of cohabiting dogs and some of the dimensions of their attachment bond with their owner. With further elaboration, rank assessment of dogs in multi-dog establishments could have applied relevance in the case of behavioural counseling or adoption by new owners. Since social rank may change with the transformation of the social structure among the cohabiting dogs (i.e., with the arrival of new dogs, the aging of the dogs), it would be interesting to examine how attachment behaviour of the same dog changes over time. According to our current results, age influences some elements of the attachment complex; however, the effect of age was not possible to investigate systematically in this study. Future research should focus on longitudinal studies in which possible changes in attachment could be examined along with aging. Another relevant future research direction would be the investigation of the effect of social dynamics on the attachment complex in larger cohabiting dog groups, where the owner has not only two, but several dogs.

## Figures and Tables

**Figure 1 animals-15-01916-f001:**
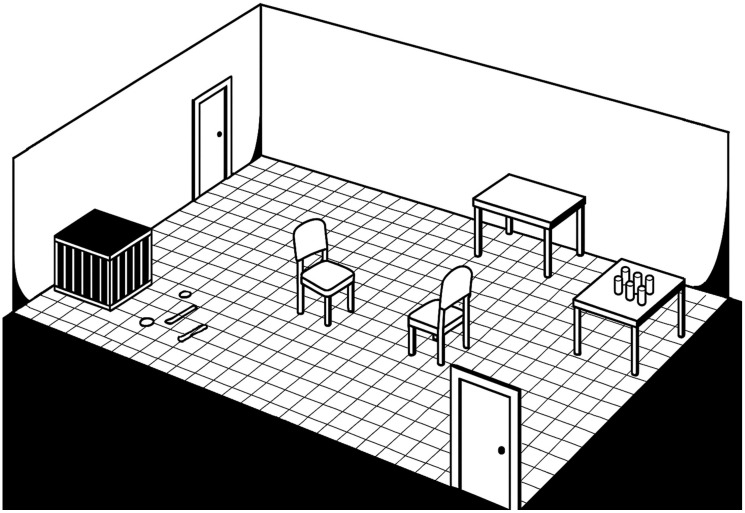
Schematic drawing of the experimental setting.

**Figure 2 animals-15-01916-f002:**
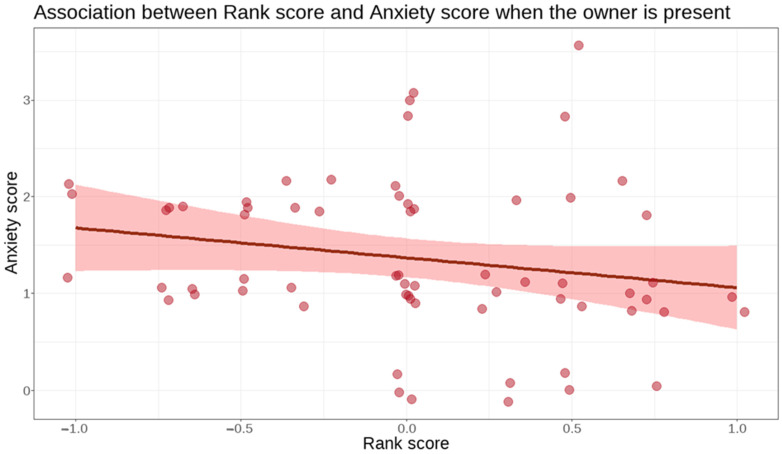
Negative association between rank score from the DRA-Q and anxiety score on the test: dogs with higher rank scores showed less anxious behaviours when the owner was present than lower-scoring dogs. Red line = Fit; shaded band = 95% confidence limits.

**Figure 3 animals-15-01916-f003:**
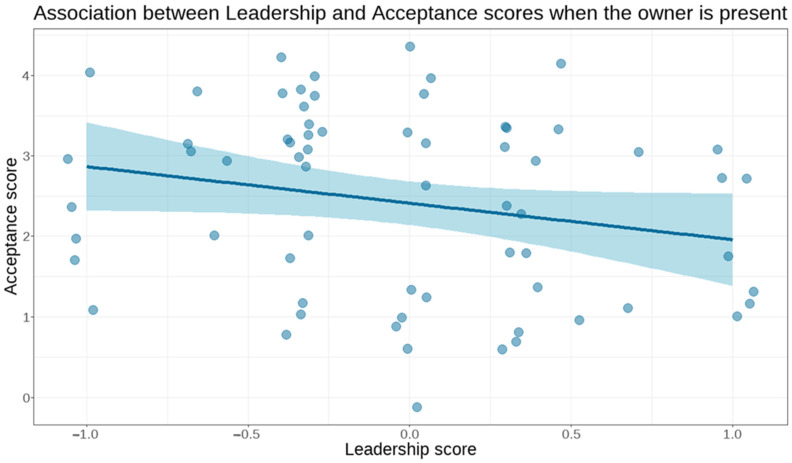
Negative association between leadership score from the DRA-Q and acceptance score on the test: dogs with higher leadership scores accepted the stranger more when the owner was present than lower-scoring dogs. Blue line = Fit; shaded band = 95% confidence limits.

**Table 1 animals-15-01916-t001:** Questions from the DRA-Q based on [35]. Explanation for rank subscore categories: “Leadership” = when one of the dogs initiates an action or takes the lead in a joint activity; “Formal” = active, non-agonistic display of subordinance/dominance; and “Agonistic” = behaviours that are related to winning in a competitive context.

Question	This Dog	One of My Other Dogs	Neither/Depends on the Situation	I Do Not Know/Not Applicable	Rank Subscore Categories
When a stranger comes to the house, which dog starts to bark first (or if they start to bark together, which dog barks more or longer)?	1	−1	0	N/A	leadership
Which dog licks more often the other dog’s mouth?	−1	1	0	N/A	formal
If the dogs receive food at the same time and at the same spot, which dog starts to eat first or eats the other dog’s food?	1	−1	0	N/A	agonistic
If the dogs start to fight, which dog usually wins?	1	−1	0	N/A	agonistic
If they receive a special reward (e.g., a marrow bone), which dog obtains it?	1	−1	0	N/A	agonistic
Which dog goes in the front during walks?	1	−1	0	N/A	leadership
Which dog acquires the better resting place?	1	−1	0	N/A	agonistic
If your dogs are being attacked, which dog faces the threat in the front?	1	−1	0	N/A	leadership

**Table 2 animals-15-01916-t002:** The episodes of the strange situation test, with the actions performed by the owner and by the stranger, respectively. Ep = episode.

Ep	Time	Owner	Stranger
1.	0:00–0:30	sits still on the chair	absent
	0:30–1:00	carries cubes, from one table to the other, one at a time	
	1:00–1:30	sits still on the chair	
	1:30–2:00	plays with the dog	
	2:00	when the door opens sits still on the chair	enters the room, greets briefly if the dog approaches (5–10 s)
2.	2′–2:30	sits still on the chair	sits still on the chair
	2:30–3:00		carries cubes, from one table to the other, one at a time
	3:00–3:30		sits still on the chair
	3:30–4:00		plays with the dog
	4:00	leaves the leash on the chair and exits the room	when the door closes after the owner leaves, sits still on the chair
3.	4′–4:30	absent	sits still on the chair
	4:30–5:00		carries cubes, from one table to the other, one at a time
	5:00–5:30		sits still on the chair
	5:30–6:00		plays with the dog
	6:00	enters the room, starts playing with the dog; stops playing when the stranger leaves	after the owner initiates play, leaves the room
4.	6′–6:30	sits still on the chair	absent
	6:30–7:00	carries cubes, from one table to the other, one at a time	
	7:00–7:30	sits still on the chair	
	7:30–8:00	plays with the dog	
	8:00	leaves the room	
5.	8′–9:00	dog alone, owner absent	absent
	9:00–9′		enters the room, greets briefly if the dog approaches
	9′–10:00	absent	sits still on the chair
	10:00		leaves the room
6.	10′–11:00	dog alone, owner absent	absent
	11:00	enters the room	
	11′–12:00	sits still on the chair	
	12:00	sits 10 s after the door opens, then puts the leash on the dog and leaves the room with the experimenter	

**Table 3 animals-15-01916-t003:** The scoring of behaviours, according to the three main behavioural dimensions (attachment, anxiety, and acceptance). The numbers in the ‘Episode’ column refer to the corresponding episode of the strange situation test. For statistical analysis, the scores were summed separately for each behavioural factor in the case of each subject. O = owner; S = stranger; D = dog; e = end of the certain episode; and s = start of the certain episode.

Dimension	Context	Episode	Description of Behaviour	Score
**ATTACHMENT**				SST score
**Owner (O) PRESENT**	Proximity	1, 2, 4, 6	D is close to O (closest body part is within 1 m)—in most of the time when D does not explore or play	1
	BlockO-1	1	during the first block-carrying D watches OR follows O for more than half of the time	1
	LeaveO-1	2e	when O first leaves, D follows O to the door (at least within 1 m from door)	1
	EnterO-1	4s	when O first enters, D approaches O at once (in reaching distance) AND D wags tail	1
	BlockO-2	4	during the second block-carrying C/D watches OR follows O for more than half of the time	1
	LeaveO-2	4e	when O leaves the second time, D follows O to the door (at least within 1 m from door)	0.5
	EnterO-2	6	when O enters the second time, D approaches O at once (in reaching distance) AND D wags tail/jumps/spins	0.5
**Owner ABSENT**	DoorS-1	3	D stands by OR orients at O’s door (for at least 5 s—score 0.5; almost all the time—score 1) during first separation	1
	NoPlayS	3	D does not play with S, although it played with her more than 10 s in Episode 2 (in O’s presence)	1
	VocalS	3, 5	D vocalises (any occurrence, except D asking for ball from S)	0.5
	Chair	3, 5	D is for more than half of the time at the chair of O if it is not at the door	0.5
	DoorS-2	5	D stands by OR orients at O’s door (for at least 5 s) during second separation	1
	EscapeS	5	when S enters, D first tries to sneak out through the door instead of greeting S	0.5
	Door-3	6	D stands by OR orients at O’s door (for at least 5 s) during third separation	0.5
			sum	11.0
**ANXIETY**				
**Owner PRESENT**	DoorO-1	1	D stands at any door (for at least 5 s—score 1, almost all the time during sit/play—score 2)	2
	ContactO	1, 2	D seeks contact with O before the first separation from O	1
	VocalO	1, 2, 4, 6	D vocalises (except D asking for the ball or greeting the O)	1
	Passive	1, 2, 4	D does not play and does not explore for more than a few seconds (except in a pretty relaxed position)	1
	Hide	1, 2, 4, 6	D stays (hides) under/behind O’s chair for more than half of the time of the sit phases	1
	Lead	1, 2e, 4, 4e	as soon as O stands up, D approaches the door (going ahead of O) (4 × 0.5)	2
	DoorO-2	4	D stands at any door for at least 5 s	1
	DoorO-3	6	D stands at any door for at least 5 s	1
**Owner ABSENT**	SeparationS	3	at separation, D vocalises OR scratches the door, or D runs around up and down for at least 10 s	1
	Calm1	3	D plays or lies down comfortably (head down) for more than 10 s	1
	FollowS	4	D follows S to the door when she leaves	1
	Separation	5, 6	at separation, D **vocalises** OR scratches the door, or D runs around up and down for at least 10 s (2 × 0.5)	1
	Calm2	5, 6	D plays or lies down comfortably (head down) for more than 10 s when alone (in sum)	1
			sum	11.0
**ACCEPTANCE**				
**Owner PRESENT**	EnterS	1e	D approaches S when she first enters (at once, within reaching distance)	1
	GreetS	1e	when S first enters, D establishes physical contact with her AND D wags tail	1
	BlockS-1	2	during the block-carrying part, D watches or follows S for more than half of the time	1
	PlayS	2	D plays with S at least for 10 s	1
**Anytime**	RubS/ToyS	2, 3, 5	D offers a toy to S (not during play)	1
	ContactS	2, 3, 5	D seeks physical contact with S (jumps on, snuggles up to, or nudges) during the episodes	1
	AvoidS	2, 3	D avoids S during play (stands off, avoids her touch)	1
**Owner ABSENT**	PoximityS	3, 5	D stays close (closest body part is within 1 m) to S in sit phases (at least for 5 s—1, almost all the time—2)	2
	BlockS-2	3	during the block-carrying part, D watches or follows S for more than half of the time	1
	PlayS-2	3	D plays with S also during separation (a little—1, a lot—2)	2
	GreetS-2	5	when S enters second time, D approaches her (0.5) AND establishes physical contact with her while D wags the tail (1)	1
			sum	11.0

## Data Availability

The dataset used for the analyses is provided as Appendix A.

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
