# Peer review of "Hierarchy-Dependent Behaviour of Dogs in the Strange Situation Test: High-Ranking Dogs Show Less Stress and Behave Less Friendly with a Stranger in the Presence of Their Owner"

_animals, 2025, doi:10.3390/ani15131916_

Round 1
Reviewer 1 Report
Comments and Suggestions for Authors
The manuscript “High rank, low stress – hierarchy-dependent behaviour of dogs in the Strange Situation Test” studies the importance of intraspecific hierarchy on the dog-owner attachment bond. It is an interesting topic and the literature about the relationship of intraspecific dominance and interspecific sociocognitive skills in dogs is scarce. Overall, the manuscript is clear and well-written. The introduction, design and discussion are adequate.
I have a few minor suggestions.
The title does not reflect all the findings, perhaps only the second part “…hierarchy-dependent behaviour of dogs in the Strange Situation Test” represents more accurately the study's results.
Considering the importance of the novelty of the testing room for the attachment assessment, you should report how many dogs had previously visited the testing facility before this evaluation.
L 167 please clarify if the questionnaire was completed by the main owner of each dog. In addition, was the test conducted by the main owner?
Table 1, please add the definition of each category.
Fig 1 where were the cameras located?
L 218 How long was the duration of the petting.
Please, mention the name and order of the episodes in the procedure.
Is it possible that older dogs were more sociable overall?
Using the current data, could a binary categorization be made? For example dividing the dogs into high and low dominance groups using the percentiles (although I'm not sure if the sample size is sufficient). It would be interesting to analyze the effect of the groups on the attachment dimensions.
Author Response
Responses to REVIEWER 1
The manuscript “High rank, low stress – hierarchy-dependent behaviour of dogs in the Strange Situation Test” studies the importance of intraspecific hierarchy on the dog-owner attachment bond. It is an interesting topic and the literature about the relationship of intraspecific dominance and interspecific sociocognitive skills in dogs is scarce. Overall, the manuscript is clear and well-written. The introduction, design and discussion are adequate.
RESPONSE: Thank you for the supportive evaluation and for the excellent comments. We did our best to revise the manuscript accordingly. Please find our detailed answers in the following section. The changes are highlighted throughout the manuscript as well.
I have a few minor suggestions.
The title does not reflect all the findings, perhaps only the second part “…hierarchy-dependent behaviour of dogs in the Strange Situation Test” represents more accurately the study's results.
RESPONSE: We agree with the Reviewer that a more accurate title would be better. We rewrote the title, now it reads:
“Hierarchy-dependent behaviour of dogs in the Strange Situation Test: high-ranking dogs show less stress and behave less friendly with a stranger in the presence of their owner”
Considering the importance of the novelty of the testing room for the attachment assessment, you should report how many dogs had previously visited the testing facility before this evaluation.
RESPONSE: Thank you for the important comment. None of the subjects were tested in the exact same experimental room before (we added this detail to the text, lines 193-194), and it was also a requirement that the subjects’ last visit (if there was any) to the Department of Ethology had to be at least a year before we tested them in the SST.
L 167 please clarify if the questionnaire was completed by the main owner of each dog. In addition, was the test conducted by the main owner?
RESPONSE: Thank you for your insightful comment, this is an important detail. Yes, the main owner was asked to complete the questionnaire and to bring the dog for the test. We added this detail to the text (lines 211-212):
“Always the main owner of the dogs was requested to complete the questionnaire, who then also took part in the behavioural test with the dog.”
Table 1, please add the definition of each category.
RESPONSE: Definitions were added to the caption of Table 1 as the Reviewer requested.
Fig 1 where were the cameras located?
RESPONSE: The fixed 6-camera recording system was mounted on the ceiling of the testing room in such way that the entire room has been covered by them. We added this detail to the text (line 262).
L 218 How long was the duration of the petting.
RESPONSE: Thank you for the clarifying question. We elaborated the text (line 295), now it reads:
“The owner/stranger did not initiate interaction with the dog, but responded if the dog initiated the interaction, (e.g., by placing its head on the owner's knee) when it was allowed to briefly pet the dog (i.e., a few hand-strokes to the head).”
Please, mention the name and order of the episodes in the procedure.
RESPONSE: The Reviewer is right, these details are necessary for the article. A new table (Table 2) has been added to the manuscript with the requested details.
Is it possible that older dogs were more sociable overall?
RESPONSE: This is an interesting idea. What we found is that older dogs and dogs with high rank scores both showed lower anxiety scores, but while older dogs were more likely to accept the stranger, dogs with high rank scores were less friendly with the stranger. One could assume that low anxiety has a different background in old and in high-scoring dogs: older dogs can be more sociable, and the higher ranking dogs more confident in the presence of their owner. Both can result in lower anxiety.
Using the current data, could a binary categorization be made? For example dividing the dogs into high and low dominance groups using the percentiles (although I'm not sure if the sample size is sufficient). It would be interesting to analyze the effect of the groups on the attachment dimensions.
RESPONSE: thank you for this interesting suggestion. The recommended procedure could work, if in general, the higher rank scores would also indicate the dominant dogs. However, as the rank scores were calculated for the cohabiting dogs within a given household, in our full sample dominant and subordinate dogs (from different households) sometimes had the same rank score. This is why we did not formulate groups based on ‘low’ and ‘high’ rank scores.
Reviewer 2 Report
Comments and Suggestions for Authors
The manuscript explores a compelling question in ethology and canine-human bonding: the role of social hierarchy among cohabiting dogs in shaping stress responses and attachment behaviour. The study is novel in focusing on intra-household rank as a moderator of SST outcomes.
Strengths:
- Strong theoretical framing of attachment and social rank.
- Use of validated tools (SST and DRA-Q).
- Inclusion of multiple behavioural dimensions (attachment, anxiety, acceptance).
Areas for Improvement:
- Clarification of Predictions: The abstract and introduction suggest high-ranking dogs were expected to show higher stress and attachment behaviours. Yet the actual findings are the reverse. This needs more precise alignment in the framing or acknowledgement of the unexpected outcome.
- Statistical Reporting: Although GLMMs are appropriate, the rationale for choosing specific model structures (e.g., random intercepts for household) should be briefly elaborated.
- Figure Enhancement: A visual summary (e.g., bar plots or regression lines) showing the relationship between rank and each behavioural dimension would significantly improve accessibility and comprehension.
- Terminological Consistency: The term "attachment" is used both in a general and narrow sense (as one dimension of the test). Consider using more precise terms (e.g., "Attachment score") when referring to the operationalised variable.
- Discussion Depth: The paper excellently integrates its results with previous literature, but could benefit from a more critical stance on the limitations of owner-reported rank scores and the potential biases inherent in DRA-Q.
Author Response
RESPONSES to REVIEWER 2
The manuscript explores a compelling question in ethology and canine-human bonding: the role of social hierarchy among cohabiting dogs in shaping stress responses and attachment behaviour. The study is novel in focusing on intra-household rank as a moderator of SST outcomes.
RESPONSE: We thank the Reviewer for their supportive attitude and positive opinion about our work. We also found the recommendations valuable and used them for improving the manuscript. Please find our detailed responses below, as well as the changes throughout the manuscript (highlighted with the tracking function of MS Word).
Strengths:
- Strong theoretical framing of attachment and social rank.
- Use of validated tools (SST and DRA-Q).
- Inclusion of multiple behavioural dimensions (attachment, anxiety, acceptance).
Areas for Improvement:
- Clarification of Predictions: The abstract and introduction suggest high-ranking dogs were expected to show higher stress and attachment behaviours. Yet the actual findings are the reverse. This needs more precise alignment in the framing or acknowledgement of the unexpected outcome.
RESPONSE: thank you for this recommendation. We agree, there is an apparent difference between one of our predictions (higher-ranking dogs would show stronger stress/anxiety than the subordinate dogs when the owner is absent) and the results (higher-ranking dogs showed less stress). We elaborated on the discussion about this issue. We emphasize that the contradiction is less sharp if we consider that the lower anxiety in dogs with higher rank-scores was only noticeable when the owner was present in the SST. In other words, it was exactly the presence of the owner that attenuated their stress, meaning that the owner probably has a higher relevance/importance for the higher-scoring dogs (lines 656-662):
“We predicted that high-ranking dogs would show stronger stress-reactions during the SST, which includes episodes where the owner leaves the dog either alone, or in the company of an unknown person. However, we found that while higher ranking dogs showed fewer stress signs, lower ranking dogs stressed more, which at first sight contradicts our prediction. We should note that these differences were only noticeable when the owner was present, which still highlights the specific importance of the owner as a stress-reducer for the higher-ranking dog.”
- Statistical Reporting: Although GLMMs are appropriate, the rationale for choosing specific model structures (e.g., random intercepts for household) should be briefly elaborated.
RESPONSE: We added some text for clearer explanation of the reason why household was used as random factor (lines 480-482):
“We tested at least two dogs from each household. We used the ID of the shared household as a random factor to account for the cohabiting dogs’ rank scores’ non-independence, as well as their shared environment.”
- Figure Enhancement: A visual summary (e.g., bar plots or regression lines) showing the relationship between rank and each behavioural dimension would significantly improve accessibility and comprehension.
RESPONSE: The Reviewer is absolutely right. It was our oversight that we forgot including graphs that illustrate some of the main results. We have included two additional figures now. Figure 2 shows the negative association between rank score and dogs’ anxiety (when the owner was present). Figure 3 shows the negative association between leadership score and dogs’ acceptance towards the stranger.
- Terminological Consistency: The term "attachment" is used both in a general and narrow sense (as one dimension of the test). Consider using more precise terms (e.g., "Attachment score") when referring to the operationalised variable.
RESPONSE: Thank you for the suggestion, we agree. Throughput the revised text, now we use “attachment-dimension”, “anxiety-dimension”, “acceptance-dimension”, this way separating these variables from the whole attachment complex.
- Discussion Depth: The paper excellently integrates its results with previous literature, but could benefit from a more critical stance on the limitations of owner-reported rank scores and the potential biases inherent in DRA-Q.
RESPONSE: Thank you for this suggestion, we elaborated the limitations section accordingly (lines 734-739):
“Among the limitations, we should keep in mind that although the DRA-Q instrument showed consistent results with the outcome of such behavioural tests that showcased competitive and non-competitive interactions between cohabiting dogs [35], questionnaires that are based on the owners’ responses can always be the source of biased assessment of rank. Furthermore, the DRA-Q contains only one item that is connected to the ’formal dominance’ component, which weakens the conclusions regarding this aspect of dogs’ rank.”
Reviewer 3 Report
Comments and Suggestions for Authors
Dear Author,
Thank you for submitting this relevant and original manuscript. The paper explores a potentially meaningful relationship between hierarchical status in multi-dog households and dog–owner attachment, assessed through a combination of the DRA-Q questionnaire and the Strange Situation Test (SST). While the general topic is of interest and contributes to a growing body of literature on dog cognition and attachment, several major concerns regarding methodological clarity, conceptual assumptions, and presentation of the results need to be addressed to support the strength and validity of the conclusions drawn
Major Comments
- Conceptual Clarity and Theoretical Justification
- The paper assumes that hierarchical rank in dogs is a relatively stable trait across contexts, but this assumption is not critically discussed. Rank may fluctuate based on age, reproductive status, context, and environmental changes.
I recommend that you clarify whether you consider rank to be a dynamic or stable variable, and how this assumption affects the interpretation of results.
Additionally, consider whether it would have been more ecologically valid to assess attachment in the presence of all cohabiting dogs to preserve the natural social context.
- The influence of group size is overlooked. The behavioural dynamics in two-dog households may differ significantly from larger groups.
Please address whether the number of dogs in the household was considered as a covariate, and if not, why. This is particularly important when discussing leadership or anxiety-related behaviours.
- No veterinary behavioural assessment was performed.
How do you confirm that the dogs included were behaviourally healthy and free from anxiety disorders or other behavioural pathologies that could affect attachment behaviour?
- Methodological and Protocol Clarifications
- The description of the SST protocol is confusing and lacks detail.
I strongly recommend including a clear timeline/diagram of the test phases, specifying the sequence of events, who was present (owner, stranger, both, or none), and what actions were performed.
- The use of a pre-test “bone-guarding” growl stimulus is not justified.
Why was this added to the SST? What is the theoretical rationale for adding a stressor during the warm-up? Please clarify its timing, purpose, and precedent in the literature.
- The scoring of Attachment, Anxiety, and Acceptance is underdeveloped.
Define each score more clearly. What behaviours were assessed? Why were these dimensions chosen, and how do they relate to attachment theory?
- Only 20% of videos were coded by a blind observer.
Explain why this proportion was chosen and provide inter-rater reliability measures (e.g., ICC, Cohen’s Kappa) to support consistency of coding.
- Validation of the DRA-Q
- The manuscript repeatedly refers to the DRA-Q as a “validated questionnaire,” yet the cited validation (Vékony et al.) is only partial and primarily correlational.
Please revise this terminology to reflect the true state of validation (e.g., “preliminarily validated”) and discuss its limitations, particularly regarding construct validity, inter-rater reliability, and contextual sensitivity.
Minor and Line-by-Line Suggestions
- Title
Suggested revision: Hierarchy-dependent behaviour of dogs in the Strange Situation Test
- Lines 12–13
“which is activated in mildly stressful contexts, such as the separation from the owner”
This is misleading. The dog typically seeks proximity to the owner in response to threatening or uncertain stimuli (e.g., novel humans, sudden noise), not mild separation per se.
- Line 17
“dogs with higher rank scores were less friendly with strangers”
Specify which stranger – the one present in the SST? Clarify this.
- Line 25–27
“We hypothesized that cohabiting the position…”
Reword for clarity. For example: “We hypothesized that dogs’ hierarchical status within multi-dog households is associated with variations in their attachment and dependency behaviours toward their owner.”
- Lines 25–36 (Abstract)
Consider restructuring using standard scientific abstract format:
- Hypothesis: Dogs’ social rank is associated with dog-owner attachment patterns.
- Population: 62 cohabiting dogs.
- Method: Rank assessed via DRA-Q; attachment via SST.
- Results: Higher-ranking dogs showed fewer stress behaviours in the SST but were less friendly toward strangers. Older dogs showed reduced stress and greater friendliness.
- Conclusion: Rank and its stability may influence attachment patterns among cohabiting dogs.
- Line 93
Add: Rehn, T., McGowan, R.T.S., Keeling, L.J. (2013). Evaluating the Strange Situation Procedure (SSP) to Assess the Bond between Dogs and Humans. PLoS ONE.
- Line 124–125
“The rank position of a dog does can affect…”
Correct grammar: “The rank position of a dog can affect…”
- Line 151
“No restrictions were placed on reproductive status”
This should be discussed as it may impact social behaviour, especially in multi-dog environments.
- Line 156
“We used the DRA-Q questionnaire”
Discuss limitations of the DRA-Q (see Major Comment #3).
- Line 215–217
Protocol description unclear. Provide a detailed, structured timeline indicating exactly who was in the room during each phase and what behaviours were observed.
- Line 238–242
Define the Attachment, Anxiety, and Acceptance scoring categories clearly — what is measured and why.
- Line 248
“20% of the videos…”
Justify this proportion and provide reliability statistics.
- Line 264 and Results Section
The results are very difficult to follow. Reorganize into clear subsections, each focusing on a specific predictor or dependent variable.
- Line 307
“validated questionnaire”
See Major Comment #3. Revise language.
- Line 311
“higher scoring dogs”
Clarify: scoring on which dimension? Rank? Attachment?
Conclusion
The paper addresses a relevant question in companion animal behavioural research, but in its current form, it suffers from ambiguities in methodology, insufficient detail, and overstated claims regarding the tools used. I encourage the authors to revise the manuscript extensively, especially by:
- Clarifying and justifying the protocol
- Elaborating the scoring system and statistical models
- Nuancing claims about tool validation
- Improving clarity and readability of results
Author Response
RESPONSES TO REVIEWER 3
Dear Author,
Thank you for submitting this relevant and original manuscript. The paper explores a potentially meaningful relationship between hierarchical status in multi-dog households and dog–owner attachment, assessed through a combination of the DRA-Q questionnaire and the Strange Situation Test (SST). While the general topic is of interest and contributes to a growing body of literature on dog cognition and attachment, several major concerns regarding methodological clarity, conceptual assumptions, and presentation of the results need to be addressed to support the strength and validity of the conclusions drawn.
RESPONSE: Thank you for your supportive opinion and for the useful comments. We did our best to improve the manuscript’s content and quality (changes are tracked in the manuscript text), and we provided detailed answers to your comments in the following section.
Major Comments
- Conceptual Clarity and Theoretical Justification
- The paper assumes that hierarchical rank in dogs is a relatively stable trait across contexts, but this assumption is not critically discussed. Rank may fluctuate based on age, reproductive status, context, and environmental changes.
I recommend that you clarify whether you consider rank to be a dynamic or stable variable, and how this assumption affects the interpretation of results.
RESPONSE: Thank you for the insightful comment. By using the 8-question DRA-Q instrument, we attempted to assess dogs’ rank as a stable variable, which overarches multiple situational scenarios. This is manifested in the three subranks (Agonistic, Formal and Leadership-rank). In theory, it is possible that a dog has higher rank-score in one of the subranks and a lower score in another, but these ambiguities are evened out by using the holistic rank score. The complex association between dogs’ rank score and their personality traits (Vékony, K., Prónik, F., & Pongrácz, P. (2022). Personalized dominance–a questionnaire-based analysis of the associations among personality traits and social rank of companion dogs. Applied Animal Behaviour Science, 247, 105544.) also ensures the stability of dogs’ rank in the hierarchy even if there are temporal disturbances in the group (arrival of new dog, departure or sickness of a dog etc.). We added a longer section about this to the Discussion (lines 614-631):
“Here we should emphasize that throughout our research, we considered dogs’ rank score as a relatively stable attribute of the cohabiting dogs at the time of the testing. Although the position of dogs in the hierarchy is secured through dynamic interactions, and the ‘holistic’ rank consists of various components (‘agonistic’, ‘formal’, ‘leadership’), the multi-question DRA-Q instrument provides the opportunity for a reliable and biologically relevant assessment of the structure of an established rank-system among cohabiting dogs [35]. While environmental and demographic factors may influence the changes in the individual dog’s rank, in our investigation we tested adult subjects who spent long enough time in mutual cohabitance for the development of stable rank conditions. The complex interrelationship between dogs’ rank and their personality traits [38] could also provide a relative stability to dogs’ position in the hierarchy, protecting it from temporary fluctuations of strength and motivation, caused by, for example, sickness or environmental stress.”
Additionally, consider whether it would have been more ecologically valid to assess attachment in the presence of all cohabiting dogs to preserve the natural social context.
RESPONSE: This suggestion is interesting, because indeed, the dogs could influence each other’s reactions to the various stimuli across the episodes of the Strange Situation Test. However, one of the core elements of the SST is that the subject must be the sole participant (with the attachment figure, aka the owner). This allows us to assess the specific behavioral parameters that are characteristic to the attachment complex.
It is interesting to note that although sporadically, it has been already tested whether the presence of a cohabitant dog could have a similar effect in the SST as the caregiver/owner does. Authors did not find any convincing signs of dog-dog attachment in these papers (Sipple, N., Thielke, L., Smith, A., Vitale, K. R., & Udell, M. A. (2021). Intraspecific and interspecific attachment between cohabitant dogs and human caregivers. Integrative and Comparative Biology, 61(1), 132-139.; Mariti, C., Carlone, B., Ricci, E., Sighieri, C., & Gazzano, A. (2014). Intraspecific attachment in adult domestic dogs (Canis familiaris): Preliminary results. Applied Animal Behaviour Science, 152, 64-72.).
The influence of group size is overlooked. The behavioural dynamics in two-dog households may differ significantly from larger groups.
RESPONSE: We agree, this is a very interesting question. The practical execution of such a study is, however, would be quite difficult, because finding a large enough sample of multidog households with a larger number of dogs available for testing is a strong limiting factor.
Please address whether the number of dogs in the household was considered as a covariate, and if not, why. This is particularly important when discussing leadership or anxiety-related behaviours.
RESPONSE: Thank you for the suggestion. We checked the data: from the 31 households that were involved in the study, 26 were two-dog, 3 households with three dogs; and 2 households with five dogs in each (these details were added to the Methods). These numbers do not allow us to run reasonable statistical tests, however, we added this idea of testing hierarchy-related behaviours in larger cohabiting dog groups to the future directions (lines 760-762).
“Another relevant future research direction would be the investigation of the effect of social dynamics on the attachment-complex in larger cohabiting dog groups, where the owner has not only two, but several dogs.”
- No veterinary behavioural assessment was performed.
How do you confirm that the dogs included were behaviourally healthy and free from anxiety disorders or other behavioural pathologies that could affect attachment behaviour?
RESPONSE: Thank you for the question. We added to the methods section that when we recruited the test subjects, we set it as a preliminary criterion that owners should refrain from entering such dogs to the test who showed problematic (strong) reactions to separation from the owner (lines 194-195). We did this mostly because of animal welfare reasons, because otherwise separation behavioral problems do not affect the attachment complex in dogs (i.e., dogs with separation problems would still show exclusive attachment bond with the owner).
- Methodological and Protocol Clarifications
- The description of the SST protocol is confusing and lacks detail.
I strongly recommend including a clear timeline/diagram of the test phases, specifying the sequence of events, who was present (owner, stranger, both, or none), and what actions were performed.
RESPONSE: Thank you for the request, the suggested details were displayed in a new table (Table 2) and added to the manuscript.
- The use of a pre-test “bone-guarding” growl stimulus is not justified.
Why was this added to the SST? What is the theoretical rationale for adding a stressor during the warm-up? Please clarify its timing, purpose, and precedent in the literature.
RESPONSE: Thank you for the request for further details. The inclusion of an extra (mild) stressor during the warm-up phase was necessary some years ago (Lenkei et al., 2021), when scientists wanted to maintain the applicability of the original version of SST for dogs, introduced by Topál et al. (1998) some decades ago. This modification was initiated by the observation that more recently the dogs were less stressed in the SST procedure compared to the original studies. The average dog owners’ attitude has changed and the number of dogs that are kept only in the backyards has decreased (while in the late 1990s it was more common practice among the dog owners who volunteered for the tests. On the other hand, more and more dogs have been frequently taken to other places besides their homes and habituated to strangers. Consequently, the unfamiliar environment of the SST is probably no longer as stressful for all dogs as it used to be. To reach the moderate level of experienced stress, which is the main causative feature of the SST (Ainsworth and Wittig, 1969), an 8-second-long dog growl (from a so-called food-guarding context, see Faragó et al., 2010) was played to the subjects during the warm-up phase. Growling is a vocalisation evoked in agonistic situations in canines and it was found that even played back growling sounds cause increased cortisol level and also behavioural reactions, such as avoidance, in dogs (e.g.: Wood et al., 2014; Faragó et al., 2010). We opted to use these low-intensity agonistic dog vocalizations because, besides being moderately stressful for the subjects, they were otherwise not connected to the separation episodes of the SST or to the unfamiliar person acting as the ‘stranger’ in the SST.
We added a brief resume of this explanation to the manuscript as well (lines 244-249):
“This additional stressor became necessary after more than 20 years of the original study on dogs’ attachment behaviour by Topál and colleagues [28], when researchers noticed that companion dogs did not show the optimal low-level stress in the SST anymore [31]. The assumed reason for this change in the dogs’ behaviour was the recent increased level of exposure to unknown places, persons and situations in the average companion dogs.”
- The scoring of Attachment, Anxiety, and Acceptance is underdeveloped.
Define each score more clearly. What behaviours were assessed? Why were these dimensions chosen, and how do they relate to attachment theory?
RESPONSE: The list of behavioural variables can be seen in Table 3, where we listed all the behaviours according to which scoring dimension they were used for. This scoring system was established long ago, here we followed a recent version, used by Kovács et al. (Kovács, K., Virányi, Z., Kis, A., Turcsán, B., Hudecz, Á., Marmota, M. T., ... & Topál, J. (2018). Dog-owner attachment is associated with oxytocin receptor gene polymorphisms in both parties. A comparative study on Austrian and Hungarian border collies. Frontiers in Psychology, 9, 435.). Since then, the method appeared in other publications as well, even with different animal species in the focus (e.g., Gábor, A., Pérez Fraga, P., Gácsi, M., Gerencsér, L., & Andics, A. (2024). Domestication and exposure to human social stimuli are not sufficient to trigger attachment to humans: a companion pig-dog comparative study. Scientific Reports, 14(1), 14058; Pongrácz, P., Bensaali-Nemes, F., Bánszky, N., & Dobos, P. (2025). The biological irrelevance of ‘Cattachment’–It’s time to view cats from a different perspective. Applied Animal Behaviour Science, 106641.)
We added more details of the behavioural elements that belong to the individual dimensions of the Attachment complex (lines 318-330).
- Only 20% of videos were coded by a blind observer.
Explain why this proportion was chosen and provide inter-rater reliability measures (e.g., ICC, Cohen’s Kappa) to support consistency of coding.
RESPONSE: The outcome (ICC values) of the inter-rater reliability analysis was moved from the Results chapter to the Behavioural Coding chapter, thus now it can be found where we first mention how the reliability analysis was done. Using 20% of the measures in the inter-rater reliability analysis is widely used standard procedure.
Validation of the DRA-Q
- The manuscript repeatedly refers to the DRA-Q as a “validated questionnaire,” yet the cited validation (Vékony et al.) is only partial and primarily correlational.
Please revise this terminology to reflect the true state of validation (e.g., “preliminarily validated”) and discuss its limitations, particularly regarding construct validity, inter-rater reliability, and contextual sensitivity.
RESPONSE: To show that the DRA-Q validated by independent behavioural tests, we changed the terminology across the manuscript “behaviourally validated”. The validity of the outcome (rank score for individual dogs) of this questionnaire was shown in our earlier paper (Vékony, K., & Pongrácz, P. (2024). Many faces of dominance: the manifestation of cohabiting companion dogs’ rank in competitive and non-competitive scenarios. Animal Cognition, 27(1), 12.). There we found that cohabiting family dogs with different rank scores based on the DRA-Q assessment, behaved ‘rank-appropriately’ in biologically relevant (competitive and non-competitive) scenarios.
Minor and Line-by-Line Suggestions
- Title
Suggested revision: Hierarchy-dependent behaviour of dogs in the Strange Situation Test
RESPONSE: Thank you for the suggestion. We changed the title to a more factual compared to the original one: “Hierarchy-dependent behaviour of dogs in the Strange Situation Test: high-ranking dogs show less stress and behave less friendly with strangers in the presence of their owner”
- Lines 12–13
“which is activated in mildly stressful contexts, such as the separation from the owner”
This is misleading. The dog typically seeks proximity to the owner in response to threatening or uncertain stimuli (e.g., novel humans, sudden noise), not mild separation per se. RESPONSE: Thank you for the note. We rewrote this section (lines 18-20), it reads now like this:
“Attachment is based on the asymmetric dependence of the dog on the human partner. Dogs seek the owner’s proximity when they experience threats, and more readily explore novel stimuli when their owner is present.”
- Line 17
“dogs with higher rank scores were less friendly with strangers”
Specify which stranger – the one present in the SST? Clarify this.
RESPONSE: Thank you for noticing this detail: now we use “experimenter” instead of “strangers”.
- Line 25–27
“We hypothesized that cohabiting the position…”
Reword for clarity. For example: “We hypothesized that dogs’ hierarchical status within multi-dog households is associated with variations in their attachment and dependency behaviours toward their owner.”
RESPONSE: thank you for the suggested editing, we adopted the proposed wording.
Lines 25–36 (Abstract)
Consider restructuring using standard scientific abstract format:
RESPONSE: Thank you for this thoughtful suggestion. This journal does not use the ‘structured abstract’ format; however we agree with the Reviewer that the abstract could be better organized. We therefore rewrote the hypothesis/subjects/method section in the abstract. Now it reads like this:
“We hypothesized that dogs’ hierarchical status within multi-dog household is associated with variations in their attachment and dependency behaviours toward their owner. We tested N = 62 cohabiting companion dogs from multi-dog households. The rank score of each subject was determined with a questionnaire (DRA-Q). We used the Strange Situation Test (SST) to assess the dogs’ attachment complex towards their owner.”
- Line 93
Add: Rehn, T., McGowan, R.T.S., Keeling, L.J. (2013). Evaluating the Strange Situation Procedure (SSP) to Assess the Bond between Dogs and Humans. PLoS ONE.
RESPONSE: We thank the Reviewer for the suggested reference. After reading the paper from Rehn et al. (2013) we finally decided to not use it, because in that paper the authors tested group-kept laboratory beagles who did not have ‘owner’ in that sense as companion dogs do.
- Line 124–125
“The rank position of a dog does can affect…”
Correct grammar: “The rank position of a dog can affect…”
RESPONSE: Thank you, we corrected it!
- Line 151
“No restrictions were placed on reproductive status”
This should be discussed as it may impact social behaviour, especially in multi-dog environments.
RESPONSE: Thank you for the suggestion. In our previous paper (Vékony, K., & Pongrácz, P. (2024). Many faces of dominance: the manifestation of cohabiting companion dogs’ rank in competitive and non-competitive scenarios. Animal Cognition, 27(1), 12.) we found that the dogs’ rank score showed no significant association with the sex and reproductive status of the subjects. The other reason why we did not balance the sample of participating dogs according to their sex and reproductive status, it was the difficulty in finding enough multi-dog households where the cohabiting dogs’ age was suitable for testing and the owners were willing to bring both dogs for testing to our laboratory.
- Line 156
“We used the DRA-Q questionnaire”
Discuss limitations of the DRA-Q (see Major Comment #3).
RESPONSE: We modified the wording of the text in other places where we refer to the DRA-Q, and we use now “behaviourally validated”. This shows that in our earlier paper (Vékony et al., 2024), behavioural tests were used to assess the association between rank score and cohabiting dogs’ responses in biologically relevant scenarios.
- Line 215–217
Protocol description unclear. Provide a detailed, structured timeline indicating exactly who was in the room during each phase and what behaviours were observed.
RESPONSE: We agree with the Reviewer, the description of the episodes was unfortunately forgotten to be included in the original version of the paper. Now it has been added to the text (Table 2).
Line 238–242
Define the Attachment, Anxiety, and Acceptance scoring categories clearly — what is measured and why.
RESPONSE: Table 3 contains the full list of scored behavioural elements. We elaborated the text regarding the nature of behaviours that belong to the different dimensions of the Attachment complex (lines 318-330):
“The behavioural elements in this dimension show that the dog remains in the vicinity of the owner, follows it activity within the room, escorts the owner to the door when they leave and greets the owner when they return. Importantly, when the owner is absent from the room, the dog does not play with the stranger, but vocalises, orients towards or stands in the door where the owner had left and stays near the chair where the owner was sitting. The second dimension measured ‘Anxiety’ (stress) induced by the unfamiliar environment. The behavioural elements in this dimension show that the dog behaves anxiously even when the owner is in the room (the dog attempts to leave the room, does not play with the owner). Additionally, when the owner is absent, the dog may follow even the stranger when they leave the room. The third dimension (‘Acceptance’) evaluated and scored the interactions with the stranger. The behavioural elements show that the dog plays with the stranger, independently of the presence of the owner, approaches the stranger and initiates contact with them.”
- Line 248
“20% of the videos…”
Justify this proportion and provide reliability statistics.
RESPONSE: We moved the results of the reliability statistics to where we mention the details of this analysis (lines 338-340). The method that we were following (20% of the video footage has been assessed by an independent coder) is regarded as standard. For example, the Institute of Educational Sciences (IES) (2017) specified the types of “inter-assessor agreement reporting” that a study must include in order to be included in the What Works Clearinghouse. According to IES guidelines, two independent reviewers should code a minimum of 20% of data points across all phases and cases in a rigorous study.” Wilson-Lopez, A., Minichiello, A., & Green, T. (2019, June). An inquiry into the use of intercoder reliability measures in qualitative research. In 2019 ASEE Annual Conference & Exposition.
- Line 264 and Results Section
The results are very difficult to follow. Reorganize into clear subsections, each focusing on a specific predictor or dependent variable.
RESPONSE: In the revised paper, we added numbered sub-chapters to the Results chapter. Each sub-chapter displays results that belong to one scoring dimension of the Attachment complex. We hope this will enhance clarity.
- Line 307
“validated questionnaire”
See Major Comment #3. Revise language.
RESPONSE: We use “behaviourally validated” now.
- Line 311
“higher scoring dogs”
Clarify: scoring on which dimension? Rank? Attachment?
RESPONSE: Thank you for directing our attention to this detail. We merged two sentences here, enhancing the clarity of the text (lines 603-604):
“Dogs with higher rank scores showed less stress-related behaviour. These dogs were more confident in the owners’ presence (but not in absence), than the ones with lower rank score.”
Conclusion
The paper addresses a relevant question in companion animal behavioural research, but in its current form, it suffers from ambiguities in methodology, insufficient detail, and overstated claims regarding the tools used. I encourage the authors to revise the manuscript extensively, especially by:
- Clarifying and justifying the protocol
- Elaborating the scoring system and statistical models
- Nuancing claims about tool validation
- Improving clarity and readability of results
RESPONSE: Once more, we thank the Reviewer for their efforts to help us improving our manuscript. The details of the changes we made are listed in the previous paragraphs.
Reviewer 4 Report
Comments and Suggestions for Authors
Section 2.1. How did you determine that 62 dogs was a sufficient sample size. Given that your results include a couple of non-statistically-significant trends, does a power analysis indicate that 62 dogs was a sufficiently large sample for the design of your study and hypotheses?
On line 189 you state that the growl was played through a speaker in the crate. On lines 204-205, you state that the growl was played from the experimenter's phone. Please clarify if the sound came from the speaker in the crate or from the speaker in the phone.
Section 2.3.2.1. Because dogs rely on olfaction, would it be confusing to the test dogs to hear a growl without smelling a dog? Or, did the dogs perhaps assume that there was no dog in the crate, similar to experiences the test dog might have had of hearing a dog on TV but having no dog present? Did the dogs show any signs of stress or fear when they heard the growl?
Line 220: Does "it" refer to the owner/stranger? If so, it is somewhat strange to refer to a person as "it". Would "they" or "owner/stranger" be less awkward?
Section 2.3.2.2. On line 214 you state that there were six phases, but only five actions are listed on lines 218-231. Please clarify the relation between phases and actions.
At the top of table 2, on the right-hand side is "SST date". I understand that SST stands for "strange situation test", but please clarify what "date" means in this context.
In the stranger section of table 2, PlayS and RubS/ToyS includes the letter C, similar to S (stranger), D (dog), and O (owner). The letter C is not defined in the table header. Please clarify what C refers to.
Lines 271-274. Please clarify why you are claiming that the trend is present when the owner is present but p = .061 especially when the overall trend is not statistically significant (p = .051).
Perhaps I am misunderstanding why you are using the SST. The SST measures attachment styles such as secure attachment. But you do not present the attachment style of the dogs as measured by the SST, but rather how strongly the dog is attached to the owner. It might be helpful if you clarified how the strength of attachment is related to attachment style as they do not seem to be the same thing to me. Is the SST a valid way of measuring strength of attachment, anxiety, and acceptance?
Results: Because your primary hypothesis is that the rank of the dog in the hierarchy might be related to attachment-related behaviors, I recommend that you reorganize the results section to present the rank results right after the intercoder reliability results.
Results: You have performed many inferential tests each with an alpha comparison-wise level of .05. Across the family of tests, the alpha family-wise level will likely be higher than .05. This implies that some of your results are likely Type-I errors. This may especially be true given that you sometimes interpret statistically non-significant results (p > .05) as trends. Given the alpha family-wise error rate, I recommend that you remove all discussions of "trends" and only discuss them in the discussion section as items for future research to consider.
Line 312: Your results for associations with rank are in terms of anxiety and acceptance. On line 312, you state "higher scoring dogs were more confident in the owner's presence..." Either show how "confidence" is related to anxiety and acceptance or word the discussion in terms of the measured variables.
Line 328: Would it be more appropriate to say that the "attachment complex" (which you indicates includes attachment, anxiety, and acceptance) rather than attachment per se (which did not differ between dogs with high- and low-ranks) could be "one of the most fundamental features of the dog-human relationship"? How do you know that it is "one of the most fundamental features" when you did not look at other features that might determine the dog-human relationship -- e.g. how long the dog has lived with the owner, the owner's behavioral style toward the dog (e.g. strict vs. loving), dog personality, human personality, etc. Your conclusion seems to be too extreme given the data presented.
You tend to use the terms "stress" and "anxiety" interchangeably throughout the manuscript (e.g., The title of table 2 talks about attachment, anxiety, and acceptance, but the body of the table uses the term "stress"). From a psychological perspective, the two are not synonymous. Since your method involves short-term, external stressors, "stress" seems to be a better term to use. Minimally, be consistent in the term you use.
Line 363: Writing styles change across time, but many years ago when I was learning the scientific writing style, I was taught that contractions (won't) were too informal for scientific writing. If you do not want to change it, that is fine with me.
Author Response
RESPONSES to REVIEWER 4
Section 2.1. How did you determine that 62 dogs was a sufficient sample size. Given that your results include a couple of non-statistically-significant trends, does a power analysis indicate that 62 dogs was a sufficiently large sample for the design of your study and hypotheses?
RESPONSE: We determined the desired sample size by using the equation for finite populations.
z (z-score) = 1.96 for 95% confidence level, ℇ (margin of error) = 0.05; pÌ‚ (population proportion) = 0.50. We expected that the population of suitable dogs (N) for our test would be 80. Based on previous social media subject recruiting campaigns and a reasonable time frame, the number of participants whom we can expect to invite is usually no more than 100 dogs – when we do not need multi-dog households. However in this study, we needed two cohabitant dogs from each household, therefore our expected population size was lower. The calculated sample size was N=67, which we eventually could almost reach in this study. We added the somewhat low sample size as a limiting factor to the Limitations section (lines 742-745).
“The study could have a higher sample size as well. However, finding owners who had at least two suitable cohabiting dogs available for testing was a strong limiting factor, for example because none of the dogs who earlier was tested in our testing room could participate in this experiment.”
On line 189 you state that the growl was played through a speaker in the crate. On lines 204-205, you state that the growl was played from the experimenter's phone. Please clarify if the sound came from the speaker in the crate or from the speaker in the phone.
RESPONSE: Thank you for your comment – we elaborated this part for more clarity. The sound came from the speaker, but the playback material was on the phone of the experimenter (lines 274-275):
“When the dog approached the crate, the experimenter played a growling sound from their phone via Bluetooth on the speaker in the crate.”
Section 2.3.2.1. Because dogs rely on olfaction, would it be confusing to the test dogs to hear a growl without smelling a dog? Or, did the dogs perhaps assume that there was no dog in the crate, similar to experiences the test dog might have had of hearing a dog on TV but having no dog present? Did the dogs show any signs of stress or fear when they heard the growl?
RESPONSE: We followed the method (growls played back from a crate, which was covered with a blanket) of Faragó and colleagues (Faragó, T., Pongrácz, P., Range, F., Virányi, Z., & Miklósi, Á. (2010). ‘The bone is mine’: affective and referential aspects of dog growls. Animal Behaviour, 79(4), 917-925.) In that paper, dogs had to obtain a desirable treat (large meaty bone) placed near to the crate where the playback sound was coming from. The test lasted for minutes, and we did not experience any confusion in the subjects: depending on the type of the growl, they either avoided the crate, or took the bone. In our present experiment (SST), dogs did not have any targeted interaction with the covered crate, they only walked nearby when the sound was played back. We cannot be sure whether they ‘knew’ if there was a dog inside or not, however, their usual reaction was orientation and a brief withdrawal from the crate.
Line 220: Does "it" refer to the owner/stranger? If so, it is somewhat strange to refer to a person as "it". Would "they" or "owner/stranger" be less awkward?
RESPONSE: We elaborated the wording to be clearer here (line 295):
“when it was allowed for the person to briefly pet the dog (i.e., a few hand-strokes to the head).”
Section 2.3.2.2. On line 214 you state that there were six phases, but only five actions are listed on lines 218-231. Please clarify the relation between phases and actions.
RESPONSE: Now we have added a new table (Table 2) that shows the exact choreography of each episode. The five actions do not exclusively correspond to any specific episode, they can appear in more than episode, and both the owner and the stranger could perform them.
At the top of table 2, on the right-hand side is "SST date". I understand that SST stands for "strange situation test", but please clarify what "date" means in this context.
RESPONSE: Thank you for noticing this detail. This table was designed based on the original coding sheet used for the experiment, and the “date” referred to the date of the actual test. We redesigned the table, now it is Table 3 and excluded this confusing detail.
In the stranger section of table 2, PlayS and RubS/ToyS includes the letter C, similar to S (stranger), D (dog), and O (owner). The letter C is not defined in the table header. Please clarify what C refers to.
RESPONSE: Another detail that originally referred to the original scoring sheet, which was used for not only dogs, but also cats, too. Thank you for noticing this detail, all “C” letters have been removed.
Lines 271-274. Please clarify why you are claiming that the trend is present when the owner is present but p = .061 especially when the overall trend is not statistically significant (p = .051).
RESPONSE: We only used the term “trend” when the p-value fell between 0.05 and 0.09, in other words, when the result was not significant, but it was near to the significance level. Some authors prefer using “marginally significant” in such cases, however we considered more appropriate to call these results only as “trends”.
Perhaps I am misunderstanding why you are using the SST. The SST measures attachment styles such as secure attachment. But you do not present the attachment style of the dogs as measured by the SST, but rather how strongly the dog is attached to the owner. It might be helpful if you clarified how the strength of attachment is related to attachment style as they do not seem to be the same thing to me. Is the SST a valid way of measuring strength of attachment, anxiety, and acceptance?
RESPONSE: The Reviewer is right; many publications follow the direction of the original SST (devised for the assessment of infant-mother attachment bond) and the 1st paper by Topál et al. (1998) on dogs’ attachment bond with the owner. These papers sorted the subjects to various attachment styles, beyond stating that dogs show attachment towards the owner. However, other publications, while using the SST protocol, did not assess the styles of attachment. For example: Topál, J., Gácsi, M., Miklósi, Á., Virányi, Z., Kubinyi, E., & Csányi, V. (2005). Attachment to humans: a comparative study on hand-reared wolves and differently socialized dog puppies. Animal behaviour, 70(6), 1367-1375.
Gácsi, M., Topál, J., Miklósi, Á., Dóka, A., & Csányi, V. (2001). Attachment behavior of adult dogs (Canis familiaris) living at rescue centers: forming new bonds. Journal of Comparative Psychology, 115(4), 423.
Lenkei, R., Carreiro, C., Gácsi, M., & Pongrácz, P. (2021). The relationship between functional breed selection and attachment pattern in family dogs (Canis familiaris). Applied Animal Behaviour Science, 235, 105231.
In our present experiment, we also opted for not to assess separate styles of attachment of low and high-ranking dogs as we had a limited sample size and did not have biologically relevant hypotheses for the potential rank-attachment style associations.
Results: Because your primary hypothesis is that the rank of the dog in the hierarchy might be related to attachment-related behaviors, I recommend that you reorganize the results section to present the rank results right after the intercoder reliability results.
RESPONSE: The statistical models have been constructed on the separate dimensions (attachment, stress, acceptance), thus we presented the results according to the dimensions. Following the Reviewer’s request, wherever it was relevant (stress and acceptance), now we start the reporting of the results with the association between rank score and dimension.
Results: You have performed many inferential tests each with an alpha comparison-wise level of .05. Across the family of tests, the alpha family-wise level will likely be higher than .05. This implies that some of your results are likely Type-I errors. This may especially be true given that you sometimes interpret statistically non-significant results (p > .05) as trends. Given the alpha family-wise error rate, I recommend that you remove all discussions of "trends" and only discuss them in the discussion section as items for future research to consider.
RESPONSE: We found three trend-like results (Attachment dimension/ age; Attachment dimension when owner is present/ age; Anxiety dimension when owner is absent/ age). These results (where the P-value was between 0.05 and 0.09) were not considered as significant, and they did not appear in the Discussion.
Line 312: Your results for associations with rank are in terms of anxiety and acceptance. On line 312, you state "higher scoring dogs were more confident in the owner's presence..." Either show how "confidence" is related to anxiety and acceptance or word the discussion in terms of the measured variables.
RESPONSE: We deleted “confident” and replaced it with “stressed less”.
Line 328: Would it be more appropriate to say that the "attachment complex" (which you indicates includes attachment, anxiety, and acceptance) rather than attachment per se (which did not differ between dogs with high- and low-ranks) could be "one of the most fundamental features of the dog-human relationship"? How do you know that it is "one of the most fundamental features" when you did not look at other features that might determine the dog-human relationship -- e.g. how long the dog has lived with the owner, the owner's behavioral style toward the dog (e.g. strict vs. loving), dog personality, human personality, etc. Your conclusion seems to be too extreme given the data presented.
RESPONSE: Thank you for the question. Throughout the manuscript now we differentiate better between the dimensions (attachment, stress, acceptance) and the whole attachment complex. As you have suggested, at this section of the text we changed the wording to “attachment complex”, and instead of “fundamental”, we use the word “stable” (lines 638-639). Earlier comparative research between dogs and tame wolves (e.g., Gácsi et al., 2005) showed that capacity for forming attachment bonds with humans could be one of the fundamental features of dog-human relationship in an evolutionary framework. The results of our study can be rather interpreted as an indicator of the stable nature of the dog-owner attachment bond on the level of an individual dog’s life.
You tend to use the terms "stress" and "anxiety" interchangeably throughout the manuscript (e.g., The title of table 2 talks about attachment, anxiety, and acceptance, but the body of the table uses the term "stress"). From a psychological perspective, the two are not synonymous. Since your method involves short-term, external stressors, "stress" seems to be a better term to use. Minimally, be consistent in the term you use.
RESPONSE: The Reviewer is right, we corrected the content of the table (currently it is Table 3), and use ‘Anxiety’ everywhere as the name of one of the dimensions of the Attachment complex. This also aligns with the terminology of related literature.
Line 363: Writing styles change across time, but many years ago when I was learning the scientific writing style, I was taught that contractions (won't) were too informal for scientific writing. If you do not want to change it, that is fine with me.
RESPONSE: Thank you for the careful suggestion, we fixed this.
Reviewer 5 Report
Comments and Suggestions for Authors
Congratulations to the authors!
This work is of great importance in dog shelters when evaluating animals at the time of adoption. There may be differences in the animals in the group that are more subordinate when adapting to a new environment and owner.
I suggest making the Objective of the work clearer and more direct, so that the objective can make a direct link with the Conclusion.
Item 5 basically talks about the limitations, please also speak objectively about the conclusions.
Author Response
RESPONSES To REVIEWER 5
Congratulations to the authors!
This work is of great importance in dog shelters when evaluating animals at the time of adoption. There may be differences in the animals in the group that are more subordinate when adapting to a new environment and owner.
RESPONSE: We thank the Reviewer for their supportive opinion about our work. This comment points to a very interesting, applied aspect of the rank assessment of dogs. However, we should keep it in mind that dogs’ rank score mainly has its relevance while the dog stays in its original group. In the case of adoption, unless the cohabiting dogs move to a new owner together, we can expect that the dogs will obtain different rank eventually in their new environment.
I suggest making the Objective of the work clearer and more direct, so that the objective can make a direct link with the Conclusion.
RESPONSE: We separated the hypotheses and predictions to a new sub-chapter for better clarity.
Item 5 basically talks about the limitations, please also speak objectively about the conclusions.
RESPONSE: The Reviewer is right. We added some text that both concludes our findings and puts them into an applied framework (lines 760-762). We also changed the title of Chapter 5 to “Liitations and future directions”.
“Our study is the first that showed associations between the rank of cohabiting dogs and some of the dimensions of their attachment bond with their owner. With further elaboration, rank-assessment of dogs in multi-dog establishments could have applied relevance in the case of behavioural counseling or adoption by new owners.”
Round 2
Reviewer 3 Report
Comments and Suggestions for Authors
Dear Author,
Thank you for submitting this relevant and original manuscript. The paper explores a potentially meaningful relationship between hierarchical status in multi-dog households and dog–owner attachment, assessed through a combination of the DRA-Q questionnaire and the Strange Situation Test (SST). While the general topic is of interest and contributes to a growing body of literature on dog cognition and attachment, several major concerns regarding methodological clarity, conceptual assumptions, and presentation of the results need to be addressed to support the strength and validity of the conclusions drawn.
RESPONSE: Thank you for your supportive opinion and for the useful comments. We did our best to improve the manuscript’s content and quality (changes are tracked in the manuscript text), and we provided detailed answers to your comments in the following section.
Major Comments
- Conceptual Clarity and Theoretical Justification
- The paper assumes that hierarchical rank in dogs is a relatively stable trait across contexts, but this assumption is not critically discussed. Rank may fluctuate based on age, reproductive status, context, and environmental changes.
I recommend that you clarify whether you consider rank to be a dynamic or stable variable, and how this assumption affects the interpretation of results.
RESPONSE: Thank you for the insightful comment. By using the 8-question DRA-Q instrument, we attempted to assess dogs’ rank as a stable variable, which overarches multiple situational scenarios. This is manifested in the three subranks (Agonistic, Formal and Leadership-rank). In theory, it is possible that a dog has higher rank-score in one of the subranks and a lower score in another, but these ambiguities are evened out by using the holistic rank score. The complex association between dogs’ rank score and their personality traits (Vékony, K., Prónik, F., & Pongrácz, P. (2022). Personalized dominance–a questionnaire-based analysis of the associations among personality traits and social rank of companion dogs. Applied Animal Behaviour Science, 247, 105544.) also ensures the stability of dogs’ rank in the hierarchy even if there are temporal disturbances in the group (arrival of new dog, departure or sickness of a dog etc.). We added a longer section about this to the Discussion (lines 614-631):
“Here we should emphasize that throughout our research, we considered dogs’ rank score as a relatively stable attribute of the cohabiting dogs at the time of the testing. Although the position of dogs in the hierarchy is secured through dynamic interactions, and the ‘holistic’ rank consists of various components (‘agonistic’, ‘formal’, ‘leadership’), the multi-question DRA-Q instrument provides the opportunity for a reliable and biologically relevant assessment of the structure of an established rank-system among cohabiting dogs [35]. While environmental and demographic factors may influence the changes in the individual dog’s rank, in our investigation we tested adult subjects who spent long enough time in mutual cohabitance for the development of stable rank conditions. The complex interrelationship between dogs’ rank and their personality traits [38] could also provide a relative stability to dogs’ position in the hierarchy, protecting it from temporary fluctuations of strength and motivation, caused by, for example, sickness or environmental stress.”
Reviewer: Thank you for the clarification.
Additionally, consider whether it would have been more ecologically valid to assess attachment in the presence of all cohabiting dogs to preserve the natural social context.
RESPONSE: This suggestion is interesting, because indeed, the dogs could influence each other’s reactions to the various stimuli across the episodes of the Strange Situation Test. However, one of the core elements of the SST is that the subject must be the sole participant (with the attachment figure, aka the owner). This allows us to assess the specific behavioral parameters that are characteristic to the attachment complex.
It is interesting to note that although sporadically, it has been already tested whether the presence of a cohabitant dog could have a similar effect in the SST as the caregiver/owner does. Authors did not find any convincing signs of dog-dog attachment in these papers (Sipple, N., Thielke, L., Smith, A., Vitale, K. R., & Udell, M. A. (2021). Intraspecific and interspecific attachment between cohabitant dogs and human caregivers. Integrative and Comparative Biology, 61(1), 132-139.; Mariti, C., Carlone, B., Ricci, E., Sighieri, C., & Gazzano, A. (2014). Intraspecific attachment in adult domestic dogs (Canis familiaris): Preliminary results. Applied Animal Behaviour Science, 152, 64-72.).
Reviewer: Thank you for the clarification.
The influence of group size is overlooked. The behavioural dynamics in two-dog households may differ significantly from larger groups.
RESPONSE: We agree, this is a very interesting question. The practical execution of such a study is, however, would be quite difficult, because finding a large enough sample of multidog households with a larger number of dogs available for testing is a strong limiting factor.
Reviewer: Thank you for the clarification.
Please address whether the number of dogs in the household was considered as a covariate, and if not, why. This is particularly important when discussing leadership or anxiety-related behaviours.
RESPONSE: Thank you for the suggestion. We checked the data: from the 31 households that were involved in the study, 26 were two-dog, 3 households with three dogs; and 2 households with five dogs in each (these details were added to the Methods). These numbers do not allow us to run reasonable statistical tests, however, we added this idea of testing hierarchy-related behaviours in larger cohabiting dog groups to the future directions (lines 760-762).
“Another relevant future research direction would be the investigation of the effect of social dynamics on the attachment-complex in larger cohabiting dog groups, where the owner has not only two, but several dogs.”
Reviewer: Thank you for the clarification.
- No veterinary behavioural assessment was performed.
How do you confirm that the dogs included were behaviourally healthy and free from anxiety disorders or other behavioural pathologies that could affect attachment behaviour?
RESPONSE: Thank you for the question. We added to the methods section that when we recruited the test subjects, we set it as a preliminary criterion that owners should refrain from entering such dogs to the test who showed problematic (strong) reactions to separation from the owner (lines 194-195). We did this mostly because of animal welfare reasons, because otherwise separation behavioral problems do not affect the attachment complex in dogs (i.e., dogs with separation problems would still show exclusive attachment bond with the owner).
Reviewer: I must respectfully disagree with the statement that separation-related behavioral problems do not affect the attachment system in dogs. In fact, these issues most often occur in anxious dogs, and the root cause is frequently a disturbance in attachment quality. For a detailed discussion, see: Masson, S., Bleuer-Elsner, S., Muller, G., Médam, T., Chevallier, J., Gaultier, E. (2024). Attachment Axis Disorders. In Veterinary Psychiatry of the Dog. Springer, Cham. https://doi.org/10.1007/978-3-031-53012-8_11.
Therefore, the absence of a behavioral assessment conducted by a qualified professional—ideally a Veterinary Specialist in Behavioral Medicine certified by EBVS or ABVS—should be clearly acknowledged as a limitation
- Methodological and Protocol Clarifications
- The description of the SST protocol is confusing and lacks detail.
I strongly recommend including a clear timeline/diagram of the test phases, specifying the sequence of events, who was present (owner, stranger, both, or none), and what actions were performed.
RESPONSE: Thank you for the request, the suggested details were displayed in a new table (Table 2) and added to the manuscript.
Reviewer: Thank you for the clarification — it’s much more understandable that way.
- The use of a pre-test “bone-guarding” growl stimulus is not justified.
Why was this added to the SST? What is the theoretical rationale for adding a stressor during the warm-up? Please clarify its timing, purpose, and precedent in the literature.
RESPONSE: Thank you for the request for further details. The inclusion of an extra (mild) stressor during the warm-up phase was necessary some years ago (Lenkei et al., 2021), when scientists wanted to maintain the applicability of the original version of SST for dogs, introduced by Topál et al. (1998) some decades ago. This modification was initiated by the observation that more recently the dogs were less stressed in the SST procedure compared to the original studies. The average dog owners’ attitude has changed and the number of dogs that are kept only in the backyards has decreased (while in the late 1990s it was more common practice among the dog owners who volunteered for the tests. On the other hand, more and more dogs have been frequently taken to other places besides their homes and habituated to strangers. Consequently, the unfamiliar environment of the SST is probably no longer as stressful for all dogs as it used to be. To reach the moderate level of experienced stress, which is the main causative feature of the SST (Ainsworth and Wittig, 1969), an 8-second-long dog growl (from a so-called food-guarding context, see Faragó et al., 2010) was played to the subjects during the warm-up phase. Growling is a vocalisation evoked in agonistic situations in canines and it was found that even played back growling sounds cause increased cortisol level and also behavioural reactions, such as avoidance, in dogs (e.g.: Wood et al., 2014; Faragó et al., 2010). We opted to use these low-intensity agonistic dog vocalizations because, besides being moderately stressful for the subjects, they were otherwise not connected to the separation episodes of the SST or to the unfamiliar person acting as the ‘stranger’ in the SST.
We added a brief resume of this explanation to the manuscript as well (lines 244-249):
“This additional stressor became necessary after more than 20 years of the original study on dogs’ attachment behaviour by Topál and colleagues [28], when researchers noticed that companion dogs did not show the optimal low-level stress in the SST anymore [31]. The assumed reason for this change in the dogs’ behaviour was the recent increased level of exposure to unknown places, persons and situations in the average companion dogs.”
Reviewer: Thank you for the clarification
- The scoring of Attachment, Anxiety, and Acceptance is underdeveloped.
Define each score more clearly. What behaviours were assessed? Why were these dimensions chosen, and how do they relate to attachment theory?
RESPONSE: The list of behavioural variables can be seen in Table 3, where we listed all the behaviours according to which scoring dimension they were used for. This scoring system was established long ago, here we followed a recent version, used by Kovács et al. (Kovács, K., Virányi, Z., Kis, A., Turcsán, B., Hudecz, Á., Marmota, M. T., ... & Topál, J. (2018). Dog-owner attachment is associated with oxytocin receptor gene polymorphisms in both parties. A comparative study on Austrian and Hungarian border collies. Frontiers in Psychology, 9, 435.). Since then, the method appeared in other publications as well, even with different animal species in the focus (e.g., Gábor, A., Pérez Fraga, P., Gácsi, M., Gerencsér, L., & Andics, A. (2024). Domestication and exposure to human social stimuli are not sufficient to trigger attachment to humans: a companion pig-dog comparative study. Scientific Reports, 14(1), 14058; Pongrácz, P., Bensaali-Nemes, F., Bánszky, N., & Dobos, P. (2025). The biological irrelevance of ‘Cattachment’–It’s time to view cats from a different perspective. Applied Animal Behaviour Science, 106641.)
We added more details of the behavioural elements that belong to the individual dimensions of the Attachment complex (lines 318-330).
Reviewer: I understand, however, that the terminology you use causes some confusion on my end. When you refer to “attachment,” I interpret it as a complex behavioral process, which doesn’t seem to match your description. In fact, secure attachment is, in my view, the ability to feel pleasure when close to the attachment figure and to seek that figure in times of danger, as they provide emotional relief and safety.
What you describe in paragraph 2.4 does not, to me, represent attachment per se, but rather the dog’s proximity-seeking behavior toward the owner. I would therefore suggest renaming the label or title of your variable to something like “owner proximity-seeking”.
The term “anxiety” also raises concerns for me. Unfortunately, there is still no scientific consensus on its precise meaning. For some, anxiety is a disorder linked to a dysfunctional neural circuit. For others, it is synonymous with fear. Yet others equate it with stress, which to me refers to elevated blood cortisol levels. As you can see, there is a lack of agreement.
I would therefore propose modifying your label and avoiding the term “anxiety.” A more appropriate alternative might be “distress expression” or “canine distress.” But as it stands, the term “anxiety” is problematic.
As for “acceptance,” that’s perhaps acceptable, though I would personally prefer something like “proximity-seeking toward a stranger” or “quality of interaction with the stranger.” These formulations seem clearer and more specific to me. The current terminology presents difficulties due to the lack of consensus surrounding their definitions
- Only 20% of videos were coded by a blind observer.
Explain why this proportion was chosen and provide inter-rater reliability measures (e.g., ICC, Cohen’s Kappa) to support consistency of coding.
RESPONSE: The outcome (ICC values) of the inter-rater reliability analysis was moved from the Results chapter to the Behavioural Coding chapter, thus now it can be found where we first mention how the reliability analysis was done. Using 20% of the measures in the inter-rater reliability analysis is widely used standard procedure.
Reviewer: Thank you for the clarification
Validation of the DRA-Q
- The manuscript repeatedly refers to the DRA-Q as a “validated questionnaire,” yet the cited validation (Vékony et al.) is only partial and primarily correlational.
Please revise this terminology to reflect the true state of validation (e.g., “preliminarily validated”) and discuss its limitations, particularly regarding construct validity, inter-rater reliability, and contextual sensitivity.
RESPONSE: To show that the DRA-Q validated by independent behavioural tests, we changed the terminology across the manuscript “behaviourally validated”. The validity of the outcome (rank score for individual dogs) of this questionnaire was shown in our earlier paper (Vékony, K., & Pongrácz, P. (2024). Many faces of dominance: the manifestation of cohabiting companion dogs’ rank in competitive and non-competitive scenarios. Animal Cognition, 27(1), 12.). There we found that cohabiting family dogs with different rank scores based on the DRA-Q assessment, behaved ‘rank-appropriately’ in biologically relevant (competitive and non-competitive) scenarios.
Reviewer: Thank you for the clarification
Minor and Line-by-Line Suggestions
- Title
Suggested revision: Hierarchy-dependent behaviour of dogs in the Strange Situation Test
RESPONSE: Thank you for the suggestion. We changed the title to a more factual compared to the original one: “Hierarchy-dependent behaviour of dogs in the Strange Situation Test: high-ranking dogs show less stress and behave less friendly with strangers in the presence of their owner”
Reviewer: Thank you, I’ve noticed the correction has been made.
- Lines 12–13
“which is activated in mildly stressful contexts, such as the separation from the owner”
This is misleading. The dog typically seeks proximity to the owner in response to threatening or uncertain stimuli (e.g., novel humans, sudden noise), not mild separation per se. RESPONSE: Thank you for the note. We rewrote this section (lines 18-20), it reads now like this:
“Attachment is based on the asymmetric dependence of the dog on the human partner. Dogs seek the owner’s proximity when they experience threats, and more readily explore novel stimuli when their owner is present.”
Reviewer: Thank you, I’ve noticed the correction has been made.
- Line 17
“dogs with higher rank scores were less friendly with strangers”
Specify which stranger – the one present in the SST? Clarify this.
RESPONSE: Thank you for noticing this detail: now we use “experimenter” instead of “strangers”.
Reviewer: Thank you, I’ve noticed the correction has been made.
- Line 25–27
“We hypothesized that cohabiting the position…”
Reword for clarity. For example: “We hypothesized that dogs’ hierarchical status within multi-dog households is associated with variations in their attachment and dependency behaviours toward their owner.”
RESPONSE: thank you for the suggested editing, we adopted the proposed wording.
Reviewer: Thank you, I’ve noticed the correction has been made.
Lines 25–36 (Abstract)
Consider restructuring using standard scientific abstract format:
RESPONSE: Thank you for this thoughtful suggestion. This journal does not use the ‘structured abstract’ format; however we agree with the Reviewer that the abstract could be better organized. We therefore rewrote the hypothesis/subjects/method section in the abstract. Now it reads like this:
“We hypothesized that dogs’ hierarchical status within multi-dog household is associated with variations in their attachment and dependency behaviours toward their owner. We tested N = 62 cohabiting companion dogs from multi-dog households. The rank score of each subject was determined with a questionnaire (DRA-Q). We used the Strange Situation Test (SST) to assess the dogs’ attachment complex towards their owner.”
Reviewer: Thank you, I’ve noticed the correction has been made.
- Line 93
Add: Rehn, T., McGowan, R.T.S., Keeling, L.J. (2013). Evaluating the Strange Situation Procedure (SSP) to Assess the Bond between Dogs and Humans. PLoS ONE.
RESPONSE: We thank the Reviewer for the suggested reference. After reading the paper from Rehn et al. (2013) we finally decided to not use it, because in that paper the authors tested group-kept laboratory beagles who did not have ‘owner’ in that sense as companion dogs do.
Reviewer: Thank you for the clarification
- Line 124–125
“The rank position of a dog does can affect…”
Correct grammar: “The rank position of a dog can affect…”
RESPONSE: Thank you, we corrected it!
Reviewer: Thank you, I’ve noticed the correction has been made.
- Line 151
“No restrictions were placed on reproductive status”
This should be discussed as it may impact social behaviour, especially in multi-dog environments.
RESPONSE: Thank you for the suggestion. In our previous paper (Vékony, K., & Pongrácz, P. (2024). Many faces of dominance: the manifestation of cohabiting companion dogs’ rank in competitive and non-competitive scenarios. Animal Cognition, 27(1), 12.) we found that the dogs’ rank score showed no significant association with the sex and reproductive status of the subjects. The other reason why we did not balance the sample of participating dogs according to their sex and reproductive status, it was the difficulty in finding enough multi-dog households where the cohabiting dogs’ age was suitable for testing and the owners were willing to bring both dogs for testing to our laboratory.
Reviewer: Thank you for the clarification. I suggest including this detail in the discussion section.
- Line 156
“We used the DRA-Q questionnaire”
Discuss limitations of the DRA-Q (see Major Comment #3).
RESPONSE: We modified the wording of the text in other places where we refer to the DRA-Q, and we use now “behaviourally validated”. This shows that in our earlier paper (Vékony et al., 2024), behavioural tests were used to assess the association between rank score and cohabiting dogs’ responses in biologically relevant scenarios.
Reviewer: Thank you, I’ve noticed the correction has been made.
- Line 215–217
Protocol description unclear. Provide a detailed, structured timeline indicating exactly who was in the room during each phase and what behaviours were observed.
RESPONSE: We agree with the Reviewer, the description of the episodes was unfortunately forgotten to be included in the original version of the paper. Now it has been added to the text (Table 2).
Reviewer: Thank you, I’ve noticed the correction has been made.
Line 238–242
Define the Attachment, Anxiety, and Acceptance scoring categories clearly — what is measured and why.
RESPONSE: Table 3 contains the full list of scored behavioural elements. We elaborated the text regarding the nature of behaviours that belong to the different dimensions of the Attachment complex (lines 318-330):
“The behavioural elements in this dimension show that the dog remains in the vicinity of the owner, follows it activity within the room, escorts the owner to the door when they leave and greets the owner when they return. Importantly, when the owner is absent from the room, the dog does not play with the stranger, but vocalises, orients towards or stands in the door where the owner had left and stays near the chair where the owner was sitting. The second dimension measured ‘Anxiety’ (stress) induced by the unfamiliar environment. The behavioural elements in this dimension show that the dog behaves anxiously even when the owner is in the room (the dog attempts to leave the room, does not play with the owner). Additionally, when the owner is absent, the dog may follow even the stranger when they leave the room. The third dimension (‘Acceptance’) evaluated and scored the interactions with the stranger. The behavioural elements show that the dog plays with the stranger, independently of the presence of the owner, approaches the stranger and initiates contact with them.”
Reviewer: Thank you, I’ve noticed the correction has been made. However, please see my remarks above regarding the terms attachment, anxiety, and acceptance. The terminology you use creates some confusion from my perspective.
- Line 248
“20% of the videos…”
Justify this proportion and provide reliability statistics.
RESPONSE: We moved the results of the reliability statistics to where we mention the details of this analysis (lines 338-340). The method that we were following (20% of the video footage has been assessed by an independent coder) is regarded as standard. For example, the Institute of Educational Sciences (IES) (2017) specified the types of “inter-assessor agreement reporting” that a study must include in order to be included in the What Works Clearinghouse. According to IES guidelines, two independent reviewers should code a minimum of 20% of data points across all phases and cases in a rigorous study.” Wilson-Lopez, A., Minichiello, A., & Green, T. (2019, June). An inquiry into the use of intercoder reliability measures in qualitative research. In 2019 ASEE Annual Conference & Exposition.
Reviewer: Thank you, I’ve noticed the correction has been made.
- Line 264 and Results Section
The results are very difficult to follow. Reorganize into clear subsections, each focusing on a specific predictor or dependent variable.
RESPONSE: In the revised paper, we added numbered sub-chapters to the Results chapter. Each sub-chapter displays results that belong to one scoring dimension of the Attachment complex. We hope this will enhance clarity.
Reviewer: Thank you, I’ve noticed the correction has been made.
- Line 307
“validated questionnaire”
See Major Comment #3. Revise language.
RESPONSE: We use “behaviourally validated” now.
Reviewer: Thank you, I’ve noticed the correction has been made.
- Line 311
“higher scoring dogs”
Clarify: scoring on which dimension? Rank? Attachment?
RESPONSE: Thank you for directing our attention to this detail. We merged two sentences here, enhancing the clarity of the text (lines 603-604):
“Dogs with higher rank scores showed less stress-related behaviour. These dogs were more confident in the owners’ presence (but not in absence), than the ones with lower rank score.”
Reviewer: Thank you, I’ve noticed the correction has been made.
Conclusion
The paper addresses a relevant question in companion animal behavioural research, but in its current form, it suffers from ambiguities in methodology, insufficient detail, and overstated claims regarding the tools used. I encourage the authors to revise the manuscript extensively, especially by:
- Clarifying and justifying the protocol
- Elaborating the scoring system and statistical models
- Nuancing claims about tool validation
- Improving clarity and readability of results
RESPONSE: Once more, we thank the Reviewer for their efforts to help us improving our manuscript. The details of the changes we made are listed in the previous paragraphs.
Reviewer: After these revisions, the paper is of significantly higher quality — it is much clearer and more convincing. That being said, I would like to emphasize two remaining points. First, I strongly recommend that you include in the limitations section the fact that these dogs were not evaluated by an animal health professional, particularly one specialized in mental health. This may have influenced the results, either positively or negatively. Second, I suggest reconsidering — perhaps in consultation with the editor — the terminology used for your variables. In my view, the definitions you provide do not align well with the terms used to label the variables. Again, as there is currently no scientific consensus on these concepts, these are suggestions for your consideration.
Author Response
RESPONSES TO REIEWER 3
Dear Reviewer,
We are happy to see that most of our responses to your original comments and suggestions were satisfactory. We are also thankful for your new comments, which are the testimonials of your care and insightfulness. In this response letter we will at first always show the previous exchange of thoughts regarding the particular issue, then your new comment, and finally, our current response.
- No veterinary behavioural assessment was performed.
How do you confirm that the dogs included were behaviourally healthy and free from anxiety disorders or other behavioural pathologies that could affect attachment behaviour?
ORIGINAL RESPONSE: Thank you for the question. We added to the methods section that when we recruited the test subjects, we set it as a preliminary criterion that owners should refrain from entering such dogs to the test who showed problematic (strong) reactions to separation from the owner (lines 194-195). We did this mostly because of animal welfare reasons, because otherwise separation behavioral problems do not affect the attachment complex in dogs (i.e., dogs with separation problems would still show exclusive attachment bond with the owner).
New comment from Reviewer: I must respectfully disagree with the statement that separation-related behavioral problems do not affect the attachment system in dogs. In fact, these issues most often occur in anxious dogs, and the root cause is frequently a disturbance in attachment quality. For a detailed discussion, see: Masson, S., Bleuer-Elsner, S., Muller, G., Médam, T., Chevallier, J., Gaultier, E. (2024). Attachment Axis Disorders. In Veterinary Psychiatry of the Dog. Springer, Cham. https://doi.org/10.1007/978-3-031-53012-8_11.
Therefore, the absence of a behavioral assessment conducted by a qualified professional—ideally a Veterinary Specialist in Behavioral Medicine certified by EBVS or ABVS—should be clearly acknowledged as a limitation
OUR NEW RESPONSE: We thank the Reviewer for the detailed explanation and for the suggested literature. We now fully see the reason for our earlier miscommunication. When researchers are assessing the various attachment styles (either in human infants or in dogs), the subjects’ reaction to separation becomes a crucial factor. From this point of view the Reviewer is absolutely right, because the potential separation problems of the subject can have a strong effect on their attachment style. In this study, we did not separate/analyze the styles of attachment, our approach was a more holistic one, comparing the strength of the three dimensions of the attachment complex among the dogs with low or high rank scores.
An additional side note to this issue, which is probably true for many countries: here in Hungary, unfortunately we do not have veterinarians specialized (certified) for diagnosing and treating behavioral problems…
We added to the manuscript the following text as a caveat/limitation (lines 478-483):
“In this study, we did not analyze the various attachment styles of companion dogs (e.g., [28]). Although the participating dogs did not have separation problems according to the owners, they were not assessed by a certified veterinary specialist before the testing. As separation problems were found to be associated with particular styles of attachment [48, 49], the lack of objective knowledge about our participants’ reaction to separation should be taken as a limitation.”
References to this section:
Konok, V., Marx, A., & Faragó, T. (2019). Attachment styles in dogs and their relationship with separation-related disorder–A questionnaire based clustering. Applied animal behaviour science, 213, 81-90.
Masson, S., Bleuer-Elsner, S., Muller, G., Médam, T., Chevallier, J., & Gaultier, E. (2024). Attachment Axis Disorders. In Veterinary Psychiatry of the Dog: Diagnosis and Treatment of Behavioral Disorders (pp. 407-451). Cham: Springer Nature Switzerland.
- The scoring of Attachment, Anxiety, and Acceptance is underdeveloped.
Define each score more clearly. What behaviours were assessed? Why were these dimensions chosen, and how do they relate to attachment theory?
ORIGINAL RESPONSE: The list of behavioural variables can be seen in Table 3, where we listed all the behaviours according to which scoring dimension they were used for. This scoring system was established long ago, here we followed a recent version, used by Kovács et al. (Kovács, K., Virányi, Z., Kis, A., Turcsán, B., Hudecz, Á., Marmota, M. T., ... & Topál, J. (2018). Dog-owner attachment is associated with oxytocin receptor gene polymorphisms in both parties. A comparative study on Austrian and Hungarian border collies. Frontiers in Psychology, 9, 435.). Since then, the method appeared in other publications as well, even with different animal species in the focus (e.g., Gábor, A., Pérez Fraga, P., Gácsi, M., Gerencsér, L., & Andics, A. (2024). Domestication and exposure to human social stimuli are not sufficient to trigger attachment to humans: a companion pig-dog comparative study. Scientific Reports, 14(1), 14058; Pongrácz, P., Bensaali-Nemes, F., Bánszky, N., & Dobos, P. (2025). The biological irrelevance of ‘Cattachment’–It’s time to view cats from a different perspective. Applied Animal Behaviour Science, 106641.)
We added more details of the behavioural elements that belong to the individual dimensions of the Attachment complex (lines 318-330).
New comment from the Reviewer: I understand, however, that the terminology you use causes some confusion on my end. When you refer to “attachment,” I interpret it as a complex behavioral process, which doesn’t seem to match your description. In fact, secure attachment is, in my view, the ability to feel pleasure when close to the attachment figure and to seek that figure in times of danger, as they provide emotional relief and safety.
What you describe in paragraph 2.4 does not, to me, represent attachment per se, but rather the dog’s proximity-seeking behavior toward the owner. I would therefore suggest renaming the label or title of your variable to something like “owner proximity-seeking”.
The term “anxiety” also raises concerns for me. Unfortunately, there is still no scientific consensus on its precise meaning. For some, anxiety is a disorder linked to a dysfunctional neural circuit. For others, it is synonymous with fear. Yet others equate it with stress, which to me refers to elevated blood cortisol levels. As you can see, there is a lack of agreement.
I would therefore propose modifying your label and avoiding the term “anxiety.” A more appropriate alternative might be “distress expression” or “canine distress.” But as it stands, the term “anxiety” is problematic.
As for “acceptance,” that’s perhaps acceptable, though I would personally prefer something like “proximity-seeking toward a stranger” or “quality of interaction with the stranger.” These formulations seem clearer and more specific to me. The current terminology presents difficulties due to the lack of consensus surrounding their definitions
OUR NEW RESPONSE: We are in an awkward position here, because the Reviewer proposes a well-supported and reasonable functional terminology, which we could easily adopt for the manuscript. However, this research was based on using a standard version of the canine Strange Situation Test, where data analysis and reporting must follow the already established terminology. We are simply not in the position of changing this terminology within the framework of an empirical study where we only use the SST as a ‘tool’. We agree with Reviewer that it would perhaps time to think about a better terminology for the dimensions of the attachment complex in dogs – but this goal would require an own research agenda.
We like the suggested labels by the Reviewer – therefore, as a small compromise, we added them to the description of the dimensions as explanations to the (perhaps less) appropriate original labels.
“The first assessed ‘Attachment’ (to the owner). The behavioural elements in this dimension refer to the proximity-seeking of the dog:…”
“The second dimension is called ‘Anxiety’, referring to the distress expressions of the dog, induced by the unfamiliar environment.”
“The third dimension (‘Acceptance’) measures the quality of interactions with the stranger.”
Line 151
“No restrictions were placed on reproductive status”
This should be discussed as it may impact social behaviour, especially in multi-dog environments.
RESPONSE: Thank you for the suggestion. In our previous paper (Vékony, K., & Pongrácz, P. (2024). Many faces of dominance: the manifestation of cohabiting companion dogs’ rank in competitive and non-competitive scenarios. Animal Cognition, 27(1), 12.) we found that the dogs’ rank score showed no significant association with the sex and reproductive status of the subjects. The other reason why we did not balance the sample of participating dogs according to their sex and reproductive status, it was the difficulty in finding enough multi-dog households where the cohabiting dogs’ age was suitable for testing and the owners were willing to bring both dogs for testing to our laboratory.
New comment from the Reviewer: Thank you for the clarification. I suggest including this detail in the discussion section.
OUR NEW RESPONSE: Thank you for the suggestion. We added the above text to the ‘Participants’ chapter (lines 157-163), because it provides an immediate explanation to our recruiting choices.
Line 238–242
Define the Attachment, Anxiety, and Acceptance scoring categories clearly — what is measured and why.
RESPONSE: Table 3 contains the full list of scored behavioural elements. We elaborated the text regarding the nature of behaviours that belong to the different dimensions of the Attachment complex (lines 318-330):
“The behavioural elements in this dimension show that the dog remains in the vicinity of the owner, follows it activity within the room, escorts the owner to the door when they leave and greets the owner when they return. Importantly, when the owner is absent from the room, the dog does not play with the stranger, but vocalises, orients towards or stands in the door where the owner had left and stays near the chair where the owner was sitting. The second dimension measured ‘Anxiety’ (stress) induced by the unfamiliar environment. The behavioural elements in this dimension show that the dog behaves anxiously even when the owner is in the room (the dog attempts to leave the room, does not play with the owner). Additionally, when the owner is absent, the dog may follow even the stranger when they leave the room. The third dimension (‘Acceptance’) evaluated and scored the interactions with the stranger. The behavioural elements show that the dog plays with the stranger, independently of the presence of the owner, approaches the stranger and initiates contact with them.”
Ne comment from the Reviewer: Thank you, I’ve noticed the correction has been made. However, please see my remarks above regarding the terms attachment, anxiety, and acceptance. The terminology you use creates some confusion from my perspective.
OUR NEW RESPONSE: We agree with Reviewer that the traditional terminology for the dimensions of Attachment complex are ambiguous and probably could be upgraded. We added the functional labels, suggested by the Reviewer, as explanations to the original labels in this part of the manuscript (lines 265-276).
Reviewer 4 Report
Comments and Suggestions for Authors
Thank you for addressing my concerns.
Author Response
RESPONSES TO REVIEWER 4
Thank you for addressing my concerns.
RESPONSE: Thank you for helping us to write a clearer, better paper.